# Explaining Kernel Clustering via Decision Trees

**Maximilian Fleissner**
Technical University of Munich
`fleissner@cit.tum.de`

**Leena C. Vankadara**
Amazon Research
Tübingen
`vleena@amazon.com`

**Debarghya Ghoshdastidar**
Technical University of Munich
`ghoshdas@cit.tum.de`

## Abstract

Despite the growing popularity of explainable and interpretable machine learning, there is still surprisingly limited work on inherently interpretable clustering methods. Recently, there has been a surge of interest in explaining the classic k-means algorithm, leading to efficient algorithms that approximate k-means clusters using axis-aligned decision trees. However, interpretable variants of k-means have limited applicability in practice, where more flexible clustering methods are often needed to obtain useful partitions of the data. In this work, we investigate interpretable kernel clustering, and propose algorithms that construct decision trees to approximate the partitions induced by kernel k-means, a nonlinear extension of k-means. We further build on previous work on explainable k-means and demonstrate how a suitable choice of features allows preserving interpretability without sacrificing approximation guarantees on the interpretable model.

## 1 Introduction

The increasing predictive power of machine learning has made it a popular tool in many scientific fields. Sensitive applications such as healthcare or autonomous driving however require more than just good accuracy—it is also crucial for a model's decisions to be interpretable (Tjoa & Guan, 2020; Varshney & Alemzadeh, 2017). Unfortunately, popular machine learning models are not transparent and are often referred to as "black box" approaches. The demand for explainable machine learning has led to the development of several tools over the last few years, albeit mostly for supervised learning. Methods such as LIME or Shapley values (Ribeiro et al., 2016; Lundberg & Lee, 2017) are designed to explain the prediction of any given machine learning model. However, posthoc-explainability often is unreliable and has been critized for not providing insight into the underlying model itself (Rudin, 2019). Counterfactual explanations on the other hand explicitly show the change in prediction had the input variables been different (Wachter et al., 2017), but do not lead to easily interpretable models. Hence, Rudin (2019) calls on researchers and practitioners to *"Stop explaining black box machine learning models for high stakes decisions and use interpretable models instead"*.

Decision trees are at the heart of several inherently interpretable machine learning models (Molnar, 2020; Breiman, 2017), and have recently also gained significant interest in the field of clustering (Bertsimas et al., 2018; Fraiman et al., 2013; Ghattas et al., 2017). Through recursive partitioning of the data based on individual features, they provide interpretability on a global scale by identifying the important features, while at the same time also allowing us to retrace the path of individual decisions through the tree. Despite the extensive empirical research on decision trees for clustering, Moshkovitz et al. (2020) were the first to introduce a clustering model based on decision trees that also satisfies worst-case approximation guarantees. They propose the Iterative Mistake Minimization (IMM) algorithm which approximates a given $k$-means clustering by a decision tree with $k$ leaves, where each leaf represents a cluster. The quality of the resulting interpretable clustering is measured by the *price of explainability*, defined as the ratio between the cost of the decision tree and the optimal $k$-means cost. IMM returns a decision tree with price of explainability of order $O(k^2)$, implying that the interpretable clusters achieve a cost not much worse than the optimal partition.

Various interpretable clustering solutions for the $k$-means problem with theoretical guarantees have emerged (see Appendix A for an overview). However, despite these advancements, a fundamental

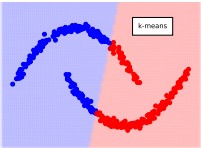 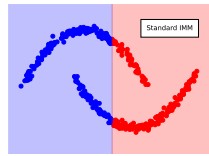 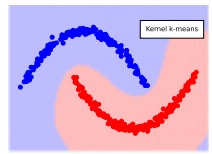 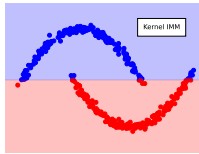

Figure 1: $k$-means does not perform well on halfmoons data that is not linearly separable, and explainable $k$-means naturally inherits its flaws. Kernel $k$-means perfectly finds the clusters and hence, its interpretable variant (proposed Kernel IMM) returns an axis-aligned decision tree with good clustering.

issue persists: the $k$-means cost is ill-suited for real-world datasets, as it fails to identify clusters that are not linearly separable. Clustering methods deployed in practice tend to be more complex, and consequently even less interpretable than standard $k$-means. The kernel $k$-means algorithm (Dhillon et al., 2004) that extends the standard $k$-means by implicitly mapping data to a reproducing kernel Hilbert space (RKHS), is particularly notable. It allows to discern clusters beyond Voronoi partitions of the input space, but this increased flexibility further diminishes the model's interpretability.

Works on interpretable kernel methods are scarce. Chau et al. (2022) propose efficient methods for computing Shapley values for kernel regression, which provide post-hoc explanations. Wu et al. (2019) enforce interpretability in kernel dimension reduction by linearly projecting the data into a low-dimensional space before computing kernel matrices—in spite of operating on fewer features, the kernel step of this approach is not interpretable. As such, the interpretability of kernel methods remains an open problem. We resolve this in the context of clustering by providing decision tree based interpretable approximations of the kernel $k$-means algorithm with provable guarantees.

**Our contributions.** While a significant body of work on decision trees for explaining $k$-means has emerged, these algorithms cannot directly be translated to the kernel setting. In particular, there are cases where explainable $k$-means does not lead to axis-aligned decision trees for *any* choice of features for the popular Gaussian kernel. We prove this in Section 3 by introducing the notions of *interpretable feature maps* and *interpretable decision trees*, which provide a theoretical characterization of the obstacles that interpretability faces in kernel clustering. Building on these insights in Section 4, we demonstrate how suitably chosen *surrogate features* can resolve this issue, and we derive a kernelized variant of the IMM algorithm (Moshkovitz et al., 2020) that preserves interpretability. These surrogate features exist for several important product kernels, including the Gaussian and Laplace kernel. Crucially, the proposed algorithm (Kernel IMM) also comes with *worst-case guarantees* on the price of explainability for a class of interpretable Taylor kernels that include the Gaussian kernel. By incorporating information encoded in the nonlinearity of kernel $k$-means, the resulting decision trees lead to significantly better results than explainable $k$-means, as Figure 1 illustrates. In Section 5, we further build on previous work (Frost et al., 2020) to derive two kernelized algorithms (Kernel ExKMC and Kernel Expand) that can be used to further refine the partitions obtained from Kernel IMM (by adding more leaves to the tree). Both can also be run on an empty tree, but do not admit worst-case bounds in this case. We conclude by demonstrating the empirical performance of the proposed methods in Section 6. The algorithms are included in Table 1.

Table 1: Overview of our algorithms ($x_i$ denotes $i$-th coordinate of point $x \in \mathbb{R}^d$)

| Algorithm | Axis-aligned cuts | Worst price of explainability |
|---|---|---|
| Kernel IMM for additive kernels | $x_i \in [\theta_1, \theta_2]$ | $O(k^2)$ (Remark 1, Appendix H) |
| Kernel IMM for interpretable Taylor kernels | $x_i \in [\theta_1, \theta_2]$ | $O(dk^2)$ (Theorem 4) |
| Kernel IMM for distance-based product kernels | $x_i \in [\theta_1, \theta_2]$ | $O(dk^2 \cdot C_{\text{data}})$ (Theorem 5) |
| Kernel ExKMC on empty tree | $x_i \gtrless \theta$ or $x_i \in [\theta_1, \theta_2]$ | Provably unbounded (Theorem 6) |
| Kernel Expand on empty tree | $x_i \gtrless \theta$ or $x_i \in [\theta_1, \theta_2]$ | Provably unbounded (Remark 2) |

## 2 BACKGROUND AND PRELIMINARIES

Consider a dataset of $n$ points, $X = \left\{ x^{(1)}, x^{(2)}, \ldots, x^{(n)} \right\} \subseteq \mathbb{R}^d$. We use $x^{(i)}$ to denote the $i$-th data point in $X$, while the notation $x_i$ denoting the $i$-th coordinate of any $x \in \mathbb{R}^d$ appears more frequently.

**$k$-means and kernel $k$-means.** For a partition of $X$ into $k$ disjoint clusters $C_1, \ldots, C_k$ with means $c_1, \ldots, c_k \in \mathbb{R}^d$, the $k$-means cost of the partition is defined as

$$cost(C_1, \ldots, C_k) = \sum_{l=1}^{k} \sum_{x \in C_l} \| x - c_l \|^2 \,. \tag{1}$$

The optimal $k$-means cost of $X$ is the minimum achievable cost, over all partitions $C_1, \ldots, C_k$. We define $\mathcal{M} = \{c_1, \ldots, c_k\}$ as the set of cluster centers obtained from some algorithm designed to minimize the $k$-means cost (1). Since finding the optimal set of centers is NP-hard (Dasgupta, 2008), in practice $\mathcal{M}$ may not necessarily consist of the $k$ optimal cluster centers. Kernel $k$-means clustering (Dhillon et al., 2004) replaces the squared Euclidean distance in (1) by a kernel-based distance. A kernel is a symmetric, positive definite function $K : \mathbb{R}^d \times \mathbb{R}^d \to \mathbb{R}$. Every kernel is implicitly associated with a feature space $\mathcal{H}$, called the reproducing kernel Hilbert space (RKHS), and a feature map $\psi : \mathbb{R}^d \to \mathcal{H}$ satisfying $K(x, y) = \langle \psi(x), \psi(y) \rangle$ for all $x, y \in \mathbb{R}^d$. For algorithms that only require knowledge of inner products between points $\psi(x)$ and $\psi(y)$, there is no need to compute the feature map explicitly, but one may instead use the kernel $K(x, y)$ to evaluate $\langle \psi(x), \psi(y) \rangle$. Thus the (implicit) feature map $\psi$ transforms the data non-linearly to a high dimensional space that provides additional flexibility in the model, without increasing the computational complexity. This is commonly known as the kernel trick. Given data $X \subseteq \mathbb{R}^d$ and a kernel $K$ on $\mathbb{R}^d$, kernel $k$-means attempts to find a partitioning $\mathcal{C} = \{C_1, ..., C_k\}$ of $X$ that minimizes

$$cost(C_1, ..., C_k) = \sum_{l=1}^{k} \sum_{x \in C_l} \| \psi(x) - c_l \|^2 \tag{2}$$

where the centers $c_1, \ldots, c_k$ are the means of each cluster in the feature space $\mathcal{H}$. Appendix B shows how the cost (2) is expressed in terms of kernel $K$ and also includes the kernel $k$-means algorithm.

**Explainable $k$-means.** The goal of explainable $k$-means clustering is to approximate a given partition $C_1, \ldots, C_k$ of $X \subseteq \mathbb{R}^d$, with centers $\mathcal{M}$, by an inherently interpretable model. Axis-aligned decision trees with $k$ leaves have emerged as a natural choice for this purpose, where every leaf of the tree $T$ corresponds to one cluster. Intuitively, one could think that the tree can simply be obtained by using a supervised learning algorithm with the cluster assignments as labels. However, Moshkovitz et al. (2020, Section 3) show that such algorithms can have arbitrarily bad worst-case approximation guarantees on the cost of the clustering induced by the decision tree, which we denote by $cost(T, X)$. This observation has sparked the development of more sophisticated approaches (Moshkovitz et al., 2020; Esfandiari et al., 2022; Charikar & Hu, 2022; Gamlath et al., 2021; Laber & Murtinho, 2021), all of which construct the tree $T$ in a top-down manner: At every internal node $u$, an axis-aligned cut $x_i \gtrless \theta$ partitions the data at node $u$, while ensuring that every cluster center $c_l \in \mathcal{M}$ ends up in exactly one leaf of $T$. This last requirement is crucial to obtain approximation guarantees. While several works restrict $T$ to have exactly $k$ leaves, few also allow more than $k$ leaves for better approximation (Makarychev & Shan, 2022; Frost et al., 2020). Appendix A provides a review of works on explainable $k$-means, while Appendix C gives a more detailed description of the Iterative Mistake Minimization (IMM) algorithm (Moshkovitz et al., 2020) that we build on in Section 4.

## 3 EXPLAINABLE KERNEL K-MEANS VIA CUTS ON FEATURE MAPS

The objective of this paper is to present an inherently interpretable model that approximates the kernel $k$-means algorithm. In this section, we make the first steps towards this goal. Standard supervised approaches for learning decision trees are questionable since they, as discussed earlier, can be arbitrarily bad for approximating even the standard $k$-means clustering. On the other hand, the aforementioned iterative decision tree construction admits good worst case guarantees in the context of $k$-means. Hence, a natural starting point would be to **kernelize** it. Unfortunately however,

the method heavily relies on explicitly characterizing the cluster centers. In kernel clustering, these cluster centers $c_1, \ldots, c_k$ reside in the RKHS $\mathcal{H}$ instead of $\mathbb{R}^d$, and typically do not have a pre-image in the input space $\mathbb{R}^d$. **Thus, operating on centers necessitates explicit computation of the feature maps, which is in contrast to the typical use of the kernel trick**.

A naïve kernelization of existing algorithms would hence aim to operate in $\mathcal{H}$ and find axis-aligned cuts using the coordinates of the projection $\psi(x)$. This approach has two fundamental limitations. *For one, it requires an explicit characterization of the RKHS $\mathcal{H}$ or the feature map $\psi$*—the two famously implicit entities in the kernel trick. Explicit characterizations of $\mathcal{H}$ or $\psi$ however are not known for most kernels, and even if they are known, the dimension of $\mathcal{H}$ could be infinite (Steinwart et al., 2006), making it practically impossible to iterate over all axes. This problem does not pose a practical concern when clustering a finite set of points. Any kernel matrix $\boldsymbol{K} \in \mathbb{R}^{n \times n}$, computed on data $X = \left\{ x^{(1)}, \ldots, x^{(n)} \right\}$, can be decomposed as $\boldsymbol{K} = \boldsymbol{\Phi}\boldsymbol{\Phi}^\top$ for some $\boldsymbol{\Phi} = \left[ \phi_1(x^{(j)}) \ldots \phi_D(x^{(j)}) \right]_{j=1}^n \in \mathbb{R}^{n \times D}$ through Cholesky or eigendecomposition, providing a data-dependent feature map that we denote as $\phi : X \to \mathbb{R}^D$.

*The second concern is how to translate axis-aligned cuts $\phi_j(x) \gtrless \theta$ back to axis-aligned cuts in $\mathbb{R}^d$*. Here, $\phi_j$ denotes the $j$-th coordinate of $\phi : X \to \mathbb{R}^D$. This issue is critically important in the context of interpretability, and can be decomposed into two principle concerns: Firstly, if $\phi_j(x)$ is a function of more than one coordinate $x_i$, then a cut $\phi_j(x) \gtrless \theta$ cannot correspond to an axis-aligned partition of $\mathbb{R}^d$. Secondly, even if $\phi_j(x)$ depends on exactly one coordinate $x_j$, the cut $\phi_j(x) \gtrless \theta$ cannot correspond to a cut $x_j \gtrless \theta'$ unless $\phi_j$ is monotonic.

These conditions are not trivial and, as we show below, the popular Gaussian kernel does not satisfy either of the two conditions, whereas some additive kernels satisfy both. To formally show this, we introduce the notion of interpretable feature maps.

**Definition 1.** *(Interpretable feature maps) Let $\phi : X \to \mathbb{R}^D$ be a feature map defined on the dataset $X \subseteq \mathbb{R}^d$ such that $K(x, y) = \langle \phi(x), \phi(y) \rangle = \sum_{j=1}^D \phi_j(x)\phi_j(y)$ for all $x, y \in X$. We say that the feature map $\phi = (\phi_1, \ldots, \phi_D)$ is interpretable if each $\phi_j(x)$ depends exactly on one coordinate of $x$.*

The following result shows that no feature map for the Gaussian kernel can be interpretable, even if we allow the feature maps to be *data-dependent*.

**Theorem 1.** *(**The Gaussian kernel cannot have interpretable feature maps**) Consider the Gaussian kernel $K(x, y) = e^{-\gamma \|x-y\|_2^2}$ in $d > 1$ dimensions. There exists a dataset $X$ such that for any feature map $\phi : X \to \mathbb{R}^D$ satisfying $\langle \phi(x), \phi(y) \rangle = K(x, y)$ for all $x, y \in X$, there exists some $j \in [D]$ such that $\phi_j$ depends on more than just one input dimension of $x \in X$.*

The short proof is given in Appendix D.1. In Section 4, we resolve the inherent non-interpretability of feature maps for the Gaussian kernel by resorting to suitably chosen *surrogate features*. However, this still leaves the concern about monotonicity of $\phi_j$. This too cannot hold for the Gaussian kernel.

**Theorem 2.** *(**The Gaussian kernel does not admit monotonic features**) Consider the one-dimensional Gaussian kernel $K(x, y) = e^{-\gamma |x-y|^2}$. Then, there exists a dataset $X \subseteq \mathbb{R}$ such that for any feature map $\phi : X \to \mathbb{R}^D$ there exists a component $\phi_j, j \in [D]$ that is not monotonic.*

The proof is included in Appendix D.2. Theorem 2 suggests that the existing notion of interpretability, specified through threshold cuts $x_i \gtrless \theta$, is too restrictive for deriving interpretable decision trees for kernels. Hence, we propose to expand the definition of interpretability. Although the partitions at each node should still align with the axes of $\mathbb{R}^d$ (this ensures that the important features can be identified), we allow the partition at the node to be any interval and its complement.

**Definition 2.** *(Interpretable decision trees) Consider a decision tree $T$ that partitions a dataset $X \subset \mathbb{R}^d$ into $k$ leaves. We call $T$ an interpretable decision tree if at every node $u \in T$, there exists some $i \in [d], \theta_1 < \theta_2 \in \mathbb{R}$ such that the data $X^u$ that arrives at $u$ is split into two disjoint subsets*

$$X_L^u = \{ x \in X^u \ : \ x_i \in [\theta_1, \theta_2] \} \quad and \quad X_R^u = \{ x \in X^u \ : \ x_i \notin [\theta_1, \theta_2] \}. \tag{3}$$

To conclude this section, we remark that several important additive kernels do in fact admit feature maps that give rise to interpretable decision trees.

**Remark 1.** *(**Certain additive kernels have interpretable and monotonic features**) Additive kernels on $\mathbb{R}^d$ refer to kernels that can be decomposed as $K(x, y) = \sum_{i=1}^d K_i(x, y)$, where each $K_i$ is*

---

**Algorithm 1** Kernel IMM for interpretable Taylor, or distance-based product kernels

---

**Input:** Data $X$, integer $k$, kernel $K$ with surrogate feature map $\phi$. For interpretable Taylor kernels see Definition 3, for distance-based kernels see Equation (4).
**Output:** Interpretable decision tree $T$ that partitions $X$ into $k$ leaves
    Get reference clustering $C_1, \ldots, C_k \leftarrow$ KERNEL $k$-MEANS$(X, K, k)$
    Compute surrogate features $\mathbf{\Phi} = \begin{bmatrix} \phi_1(x) \ldots \phi_d(x) \end{bmatrix}$ and centers $\mathcal{M} = \{c_1, \ldots, c_k\}$ under $\phi$.
    Construct a decision tree $T \leftarrow$ IMM$(\mathbf{\Phi}, y, \mathcal{M})$, where $y$ denotes the cluster assignments.
    Translate threshold cuts of $T$ back to a decision tree on $X$.

---

*a kernel on $\mathbb{R}$. Additive kernels are commonly used for comparing histograms, e.g. in computer vision (Maji et al., 2012; Vedaldi & Zisserman, 2012). In Appendix H, we consider three popular additive kernels and derive feature maps $\phi$ for which any threshold cut $\phi_j(x) \gtrless \theta$ translates to an axis-aligned cut $x_j \in [\theta_1, \theta_2]$. Hence, one may run IMM on the map $\phi$ and directly translate the tree from the feature space to an interpretable decision tree.*

## 4 KERNEL ITERATIVE MISTAKE MINIMIZATION (KERNEL IMM)

In this section, we present an interpretable kernel $k$-means algorithm that is applicable to a wide range of bounded product kernels, which include the Gaussian kernel $K(x, y) = e^{-\gamma \|x-y\|_2^2}$ as well as other distance-based product kernels such as the Laplace kernel $K(x, y) = e^{-\gamma \|x-y\|_1}$. We focus on the aforementioned idea of constructing interpretable decision trees via axis-aligned cuts of suitably chosen feature maps $\phi$, thereby leveraging existing approximation guarantees. The base interpretable $k$-means method that we "kernelize" is the IMM algorithm, primarily due to its simpler analysis. We believe the same construction can also leverage worst-case approximation guarantees for other explainable $k$-means algorithms (with tighter guarantees). The proposed Kernel IMM algorithm is described in Algorithm 1, and illustrated in Figure 2.

**Surrogate features.** Bounded product kernels on $\mathbb{R}^d$ can be expressed as $K(x, y) = \prod_{i=1}^d K_i(x, y)$, where each $K_i$ is a kernel on $\mathbb{R}$, operating on the $i$-th coordinate of data. We assume that $K_i(x, y) \leq K_i(x, x) = 1$ for all $x, y \in \mathbb{R}^d$. The RKHS of a product kernel $K$ is given by the tensorization $\mathcal{H} = \mathcal{H}_1 \otimes \ldots \otimes \mathcal{H}_d$, where $\mathcal{H}_i$ is the RKHS of kernel $K_i$. The implicit feature map $\psi : \mathbb{R}^d \to \mathcal{H}$ associated with the product kernel $K$ can be related to the feature maps $\psi_i : \mathbb{R} \to \mathcal{H}_i$ of the $d$ individual kernels $K_i$ via $\langle \psi(x), \psi(y) \rangle = \prod_{i=1}^d \langle \psi_i(x_i), \psi_i(y_i) \rangle$. In order to preserve axis-aligned structures, we propose to use a surrogate feature map $\phi = (\phi_1, \ldots, \phi_d)$, where each $\phi_i$ is a finite-dimensional approximation to $\psi_i$. Notably, $\phi$ does not approximate a map to $\mathcal{H}$ but instead to the Cartesian space $\mathcal{H}_1 \times \ldots \times \mathcal{H}_d$, and $\langle \phi(x), \phi(y) \rangle \approx \sum_{i=1}^d \langle \psi_i(x_i), \psi_i(y_i) \rangle \neq \langle \psi(x), \psi(y) \rangle$. Crucially however, we will show later that the additional cost induced by this step can be bounded. By construction, $\phi = (\phi_i)_{i=1}^d$ is an interpretable feature map, and we may compute the set of cluster centers $\mathcal{M}$ in the surrogate feature space associated with $\phi$. As described in 1, we let IMM operate on $\mathcal{M}$. Finally, interpretability of $\phi$ ensures that axis-aligned cuts of $\phi$ do in fact correspond to axis-aligned partitions of the input space $\mathbb{R}^d$. For the Gaussian kernel, our surrogate features $\phi_i$ are Taylor approximations to $\psi_i$. We also present an alternative, quite general choice for $\phi_i$ that works *for all distance-based product kernels*, but leads to data-dependent approximation guarantees.

**Interpretable Taylor kernels.** We now define a class of kernels over $\mathbb{R}$ that we refer to as interpretable Taylor kernels, and construct surrogate features for these kernels.

**Definition 3.** *(Interpretable Taylor kernels and their surrogate features) Let $K_i$ be a bounded kernel on a connected domain $\mathcal{X}_i \subseteq \mathbb{R}$, of the form $K_i(z, z') = f(z)f(z')g(zz')$ for some differentiable function $f$ and an analytic function $g$. Denote by $g^{(j)}$ the $j$-th derivative of $g$. Assume that $g^{(j)}(0) \geq 0$ for all $j \in \mathbb{N}$ and that $z^{j-1}(zf'(z) + jf(z)) = 0$ has at most one solution $z$ on $\mathcal{X}_i$. Then, we call $K_i$ an* interpretable Taylor kernel. *Furthermore, assume $K$ is a product of $d$ interpretable Taylor kernels $K_1, \ldots, K_d$, defined on a connected domain $\mathcal{X} \subset \mathbb{R}^d$. Given an integer $M$, we define the surrogate feature map for $K$ as $\phi(x) = (\phi_i(x_i))_{i=1}^d$, where each $\phi_i : \mathbb{R} \to \mathbb{R}^{M+1}$ is given by*
$$\phi_i(z) = (\phi_{i,j}(z))_{j=0}^M \text{ where we define } \phi_{i,j}(z) = z^j f(z) \sqrt{g^{(j)}(0)/j!} \text{ for all } j \leq M.$$

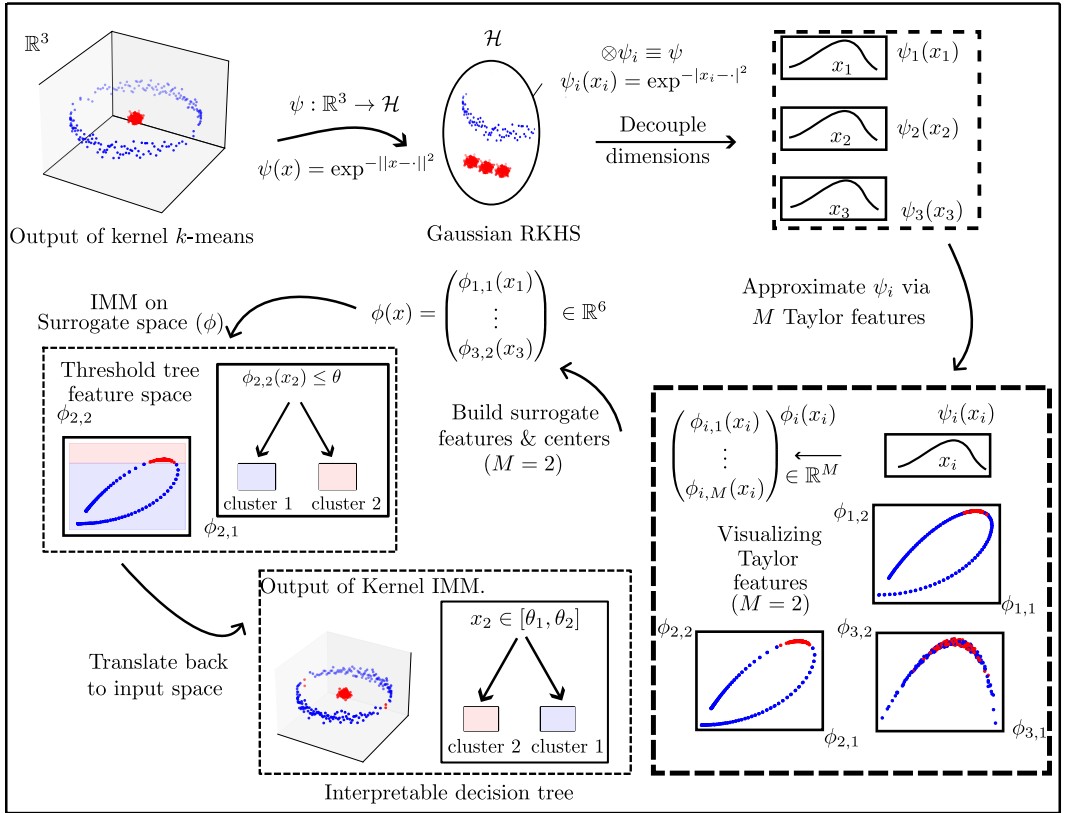

Figure 2: A schematic of Kernel IMM for Interpretable Taylor kernels.

The Gaussian kernel with bandwidth $\gamma > 0$ is a product of interpretable Taylor kernels, because with $g(z) = \exp(2z)$ and $f(z) = \exp(-\gamma z^2)$, $zf'(z) + jf(z) = 0$ has at most one solution on $(0, \infty)$. This is sufficient because we can always assume data $X$ that is clustered using the Gaussian kernel to be contained in $(0, \infty)^d$, as shifting does not affect the clustering cost.

**Theorem 3.** *(**Threshold cuts in the surrogate feature space of interpretable Taylor kernels yield interpretable decision trees**) Let $K$ be product of interpretable Taylor kernels, and let $\phi$ denote its surrogate feature map. Then, a threshold cut of the form $\phi_{i,j}(x_i) \gtrless \theta$ leads to an interpretable decision tree in $\mathbb{R}^d$.*

Our choice of surrogate features doesn't only provide interpretable decision trees, but also preserves worst-case bounds on the cost. To formalize this, we generalize the price of explainability to kernels.

**Definition 4.** *(**The price of explainability**) For a set $X \subset \mathbb{R}^d$ and an interpretable decision tree $T$ as defined in 2, the price of explainability of $T$ on $X$ is defined as the ratio $p(T, X) = cost(T, X)/cost_{opt}(X)$, where $cost_{opt}(X)$ is the optimal kernel $k$-means cost. The price of explainability of the data set $X$ is given by $p(X) = \min_T p(T, X)$, where the minimum is taken over all interpretable decision trees.*

The Kernel IMM algorithm admits worst-case guarantees quite similar to the ones obtained in the linear setting, albeit with an additional factor of order $d$. This is a result of operating on the surrogate features that decouple the $d$ input dimensions (see Appendix E).

**Theorem 4.** *(**Price of explainability for interpretable Taylor kernels**) Let $C_1, \ldots, C_k$ be the clusters of a dataset $X \subset \mathbb{R}^d$ derived from kernel $k$-means, where the kernel $K$ is an interpretable Taylor kernel. Denote by $\Phi = \phi(X)$ the surrogate features for $K$ on $X$. Then, the interpretable decision tree obtained from Kernel IMM on $\Phi$ satisfies $p(T, X) = O\left(dk^2\right) + \frac{O(dk^2\delta)}{cost_{opt}(X)}$, where $\delta = \delta(M)$ depends on the order $M$ of the surrogate feature map and $\lim_{M \to \infty} \delta(M) = 0$ for any dataset $X$. Thus, by adaptively choosing $M$ such that $\delta = O(cost_{opt})$, we obtain $O(dk^2)$ bounds.*

The proof can be found in Appendix F. For interpretable Taylor kernels, Kernel IMM operates in a feature space of dimension $O(dM)$. In practice, often even low orders $M \leq 5$ of the Taylor-based surrogate features provide enough flexibility to obtain good interpretable clusterings, and hence *Kernel IMM computationally is not much more expensive than standard IMM*.

**Extension to all distance-based product kernels.** To conclude this section, we present an extension of Kernel IMM to the entire class of distance-based product kernels $K(x, y) = \prod_{i=1}^{d} h(|x_i - y_i|)$, where $h$ is a decreasing function on the positive real line, with $h(0) = 1$. Examples include the Laplace kernel, for which $h(t) = \exp(-t)$. For these kernels, we simply set

$$\phi_{i,j}(x) = K_i(x, x^{(j)}) = h(|x_i - x_i^{(j)}|) \tag{4}$$

where $x^{(j)}$ denotes the $j$th point in the dataset $X$. Because $\phi_{i,j}(x) \leq \theta$ if and only if $|x_i - x_i^{(j)}| \geq h^{-1}(\theta)$, this leads to an interpretable decision tree. However, the worst-case bounds obtained from IMM do not translate directly.

**Theorem 5.** (*Price of explainability for distance-based product kernels*) *Let $C_1, \ldots, C_k$ be the clusters of a dataset $X \subset \mathbb{R}^d$ derived from kernel $k$-means, where the kernel $K$ is a distance-based product kernel. Denote by $\mathbf{\Phi} = \phi(X)$ the surrogate features for $K$ on $X$, where $\phi_{i,j}(x) = K_i(x, x^{(j)})$. Then, the interpretable decision tree obtained from Kernel IMM on $\mathbf{\Phi}$ satisfies $p(T, X) = O\left(Cdk^2\right)$, where $C$ depends on the dataset $X$ and the kernel $K$.*

We discuss this in Appendix G. Despite the data-dependent approximation guarantee, we observe in practice that the surrogate features defined in (4) perform well, sometimes even outperforming the surrogate Taylor features of the Gaussian kernel.

## 5 GREEDY COST MINIMIZATION

While the previous two sections cover several important kernels, there may nonetheless be cases where Kernel IMM is not applicable (because no suitable surrogate features exist), or not powerful enough to approximate kernel $k$-means (due to its restriction to just $k$ leaves, as we illustrate in Figure 3). To address these issues, we propose kernelized variants of the ExKMC algorithm (Frost et al., 2020), which greedily add new leaves. This does not require computations in the feature space: Any set of reference clusters $C_1, \ldots, C_k$ obtained from kernel k-means *implicitly* comes with a set of reference centers $\mathcal{M} = \{c_1, \ldots, c_k\} \subset \mathcal{H}$. The kernel trick ensures that the distance between the feature map of any point $\psi(x) \in \mathcal{H}$ and a cluster center $c_l$ can be evaluated without explicitly computing the center as

$$\|\psi(x) - c_l\|^2 = K(x, x) + \frac{1}{|C_l|^2} \sum_{y,z \in C_l} K(y, z) - \frac{2}{|C_l|} \sum_{y \in C_l} K(x, y) \tag{5}$$

**Kernel ExKMC (Algorithm 2).** The algorithm proceeds sequentially. For every node $u$ of the decision tree, let $X^u$ denote the set of points that reach the node $u$. A threshold cut $(i, \theta)$ is chosen, separating $X^u$ into two subsets $X_L^u = X^u \cap \{x_i \leq \theta\}$ and $X_R^u = X^u \cap \{x_i > \theta\}$. The Kernel ExKMC algorithm chooses $(i, \theta)$ as the minimizer of the cost function

$$cost_{exkmc}(u, i, \theta) = \min_{j,l \in [k]} \sum_{x \in X_L^u} \|\psi(x) - c_j\|^2 + \sum_{x \in X_R^u} \|\psi(x) - c_l\|^2 \tag{6}$$

where the squared distances can be computed using (5). One reference center is chosen for each of the two child nodes of $u$. To decide on which node to split, we choose the one that maximizes the difference between the cost of not splitting, given by $\min_{j \in [k]} \sum_{x \in X^u} \|\psi(x) - c_j\|^2$, and $cost_{exkmc}(i, \theta)$.

Since all relevant computations happen in the input space, and no explicit feature map $\psi$ is needed, the algorithm is applicable to *arbitrary* kernels and can be used to refine any given partition $\hat{C}_1, \ldots, \hat{C}_p$. Unfortunately, we can prove that Kernel ExKMC on an empty tree does not admit approximation guarantees when limited to $k$ leaves.

**Theorem 6** (**ExKMC does not admit no worst-case bounds**). *For any $m \in \mathbb{N}$ there exists a data set $X \subset \mathbb{R}$ such that the price of explainability on $X$ is $p(X) = 1$, IMM attain this minimum, but ExKMC (initialized on an empty tree) constructs a decision tree $T$ with $p(T, X) \geq m$.*

K-means      Kernel k-means      Kernel IMM      Kernel ExKMC      Kernel Expand

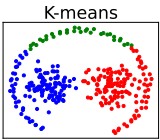 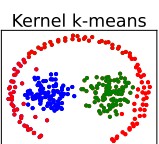 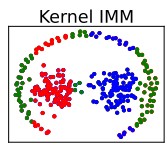 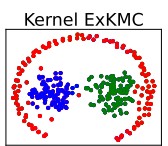 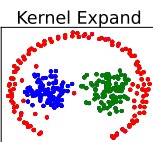

Figure 3: Standard $k$-means is ill-suited for clustering certain datasets, and this translates to explainable $k$-means (not plotted here). Kernel $k$-means recovers the ground truth well. However, Kernel IMM is restricted to 3 leaves and not powerful enough to approximate it. To resolve this, we suggest Kernel ExKMC and Kernel Expand, which extend the tree to 6 leaves.

---

**Algorithm 2** Kernel ExKMC

**Input:** Number of leaves $m \in \mathbb{N}$,
     reference clusters $C_1 \dots, C_k$,
     decision tree $T$ with leaves $\hat{C}_1, \dots, \hat{C}_p$
**Output:** Tree with $m$ leaves $\mathcal{L}$
     Initialize the set of leafs as $\mathcal{L} = \{\hat{C}_1, \dots, \hat{C}_p\}$
     **while** $|\mathcal{L}| < m$ **do**
         Choose node $X^u \in \mathcal{L}$ and a cut $(i, \theta)$ minimizing (6)
         Update $\mathcal{L}$ by replacing $X^u \in \mathcal{L}$ by its two child nodes $X_L^u, X_R^u$
     **end while**

---

**Algorithm 3** Kernel Expand

**Input:** Number of leaves $m \in \mathbb{N}$,
     reference clusters $C_1 \dots, C_k$,
     decision tree $T$ with leaves $\hat{C}_1, \dots, \hat{C}_p$
**Output:** Tree with $m$ leaves $\mathcal{L}$
     Initialize the set of leafs as $\mathcal{L} = \{\hat{C}_1, \dots, \hat{C}_p\}$
     **while** $|\mathcal{L}| < m$ **do**
         Choose data $X^u \in \mathcal{L}$ and a cut $(i, \theta)$ minimizing (7)
         Update $\mathcal{L}$ by replacing $X^u \in \mathcal{L}$ by its two child nodes $X_L^u, X_R^u$
     **end while**

---

The proof can be found in Appendix J. If Kernel ExKMC is run as a refinement of Kernel IMM (which constructs an interpretable decision tree in the sense of Definition 2 with two-sided threshold cuts) then we suggest sticking to this expanded notion of interpretability, and Kernel ExKMC at every node chooses $(i, \theta_1, \theta_2) \in [d] \times \mathbb{R}^2$ and checks whether $x_i \in [\theta_1, \theta_2]$ or not.

**Kernel Expand (Algorithm 3).** We suggest another algorithm, Kernel Expand, that also adds new leaves to an existing tree, but maximizes the purity of each leaf. This is achieved by minimizing

$$cost_{exp}(i, \theta) = \min_{j,l \in [k]} (|\{x \in X_L^u \ : \ c(x) \neq c_j\}| + |\{x \in X_R^u \ : \ c(x) \neq c_l\}|) \tag{7}$$

where $c(x)$ denotes the cluster center of $x$ according to the reference partition $\mathcal{M}$. To decide on which node to split, we again choose the one that maximizes the difference between the cost of not splitting, given by $\min_{j \in [k]} |\{x \in X^u \ : \ c(x) \neq c_j\}|$, and $cost_{expand}(i, \theta)$.

**Remark 2.** *(**Kernel Expand does not admit worst-case bounds**) In the same way as Kernel ExKMC, Kernel Expand does not admit worst-case bounds when initialized without Kernel IMM. This is obvious from the example in Section 3 of (Moshkovitz et al., 2020).*

Finally, it is important to note that while adding more leaves improves the clustering, it gradually reduces the interpretability of the tree. Thus, there is a trade-off between explainability and accuracy in Kernel ExKMC and Kernel Expand.

## 6 EXPERIMENTS

We validate our algorithms on a number of benchmark datasets, including three synthetic clustering datasets, *Pathbased*, *Aggregation* and *Flame* (Fränti & Sieranoja) and real datasets, *Iris* (Fisher, 1936) and *Wisconsin breast cancer* (Street et al., 1993). On all five datasets, we evaluate Kernel IMM as well as the greedy cost minimization algorithms from Section 5, Kernel ExKMC and Kernel Expand, both of which refine Kernel IMM by adding more leaves. All experiments reported here use either the Laplace or the Gaussian kernel. In Appendix H, we derive suitable approximate feature maps and evaluate Kernel IMM for the additive $\chi^2$ kernel, which is well-suited for clustering histograms or distributions. These results underline the usefulness of our method beyond distance-based product

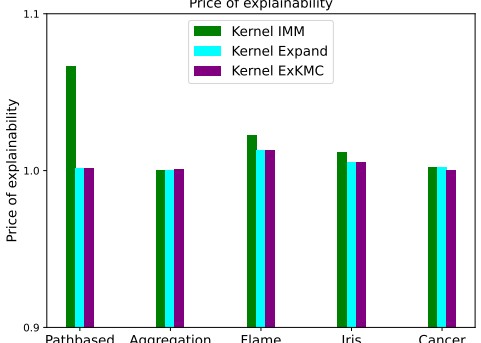 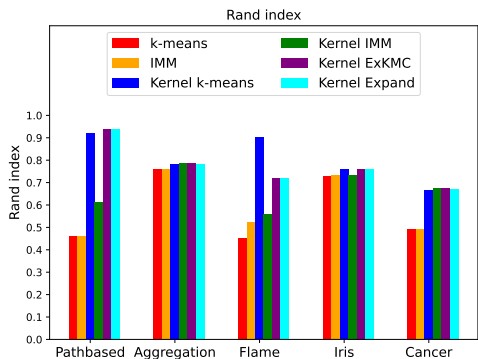

Figure 4: We verify the approximation properties of our algorithms by computing the price of explainability (left plot). We also compare the clusters obtained on $k$-means and IMM, as well as kernel $k$-means and our three algorithms to the ground truth via the Rand index (right plot).

kernels. Finally, we also compare Kernel IMM to CART, which performs similarly well. Details on all experiments can be found in Appendix I. Although Kernel IMM can lead to sub-optimal solutions, its greedy refinements mostly result in a price of explainability close to 1 (see Figure 3 for the *Pathbased* dataset). Figure 4 reports the price of explainability of the three proposed explainable kernel clustering methods (closer to 1 is better). Since the ground truth is known in all datasets considered here, Figure 4 also reports the adjusted Rand index (Pedregosa et al., 2011) that measures the agreement between the interpretable clusters and the true labels (a higher value is better). This shows that kernel $k$-means is naturally superior to $k$-means and explainable $k$-means, and one can observe that Kernel ExKMC and Kernel Expand mostly achieve Rand index as good as that of kernel $k$-means, establishing that our methods provide both interpretable and accurate clusters. In addition, Kernel IMM improves over $k$-means and IMM in recovering the ground truth.

## 7 CONCLUSION AND DISCUSSION

In this paper, we contribute to the increasing need for interpretable clustering methods by explaining kernel $k$-means using decision trees. By operating on carefully chosen features, interpretability can be preserved, while still allowing for approximation guarantees. We characterize the obstacles that interpretability faces in kernel clustering, and find that *interpretable feature maps* play a key role. These maps may also be useful to understand the interpretability of other kernel methods, which still rely almost exclusively on post-hoc explainability.

**On the price of explainability for kernels.** We observe below that the price of explainability for kernel $k$-means, as defined in Definition 4, may not be finite for arbitrary kernels.

**Theorem 7** (**Unbounded price of explainability**). *Consider either the quadratic kernel $K(x, y) = \langle x, y \rangle^2$ or the $\epsilon$-neighborhood kernel $K(x, y) = \mathbf{1}_{[0,\epsilon]}(\|x - y\|_2)$. For both, there exists a dataset $X$ such that $p(X) = \infty$.*

The above result, proved in Appendix K, is in stark contrast to standard $k$-means, where IMM always ensures a bounded price of explainability. Our experiments further show that the price of explainability is close to 1 even in cases where the interpretable clustering model is quite different from the reference partition. In fact, if the kernel $k$-means cost is $\Omega(n)$ for a dataset of size $n$ and the kernel is bounded, then the price of explainability is $O(1)$. This indicates that typical metrics should be adjusted depending on the kernel and the data. While the Rand index may be used if a ground truth is known, a fundamental question remains unsolved: *Under what conditions on the data can a decision tree achieve a good agreement with the nonlinear partitions of kernel $k$-means?* Another question is whether approximation guarantees can be improved. It is very likely that specific kernels benefit from algorithms tailored to their cost functions. Finally, investigating lower bounds (beyond the ones a linear kernel inherits from existing results on explainable $k$-means) also poses an interesting direction for future work.

**Ethics statement.**    While we do not expect negative societal impacts of our method, we remark that clustering algorithms themselves can be biased, and this can translate to its interpretable approximations.

**Reproducibility.**    We add further details on the experiments in Appendix I. In addition, all proofs for our theoretical results can be found in the appendix, and have been referenced in the main paper.

ACKNOWLEDGMENTS

This paper is supported by the DAAD programme Konrad Zuse Schools of Excellence in Artificial Intelligence, sponsored by the Federal Ministry of Education and Research, and the German Research Foundation (Research Grant GH 257/4-1).

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

## A    RELATED WORK ON EXPLAINABLE CLUSTERING

There are several papers that suggest improvements of the original IMM algorithm (Moshkovitz et al., 2020). Most rely on random threshold cuts to obtain better worst-case bounds on the price of explainability for $k$-means and $k$-medians (Makarychev & Shan, 2022; Esfandiari et al., 2022; Makarychev & Shan, 2023). As of today, the sharpest known upper bound on the price of explainability for $k$-means is $O(k \log \log k)$ (Gupta et al., 2023). It is also known that the price of explainability for $k$-means is $\Omega(k)$ (Gamlath et al., 2021), leaving a small gap. While the above works provide dimension-independent guarantees, it is also known that in low dimensions, even better guarantees can be provided (Laber & Murtinho, 2021; Charikar & Hu, 2022). A different line of research has loosened the restriction to decision trees with $k$ leaves, leading to both improved practical performance (Frost et al., 2020) and better worst-case bounds on the price of explainability (Makarychev & Shan, 2022). In addition, Laber et al. (2023) investigate shallow decision trees for explainable clustering, while Deng et al. (2023) show that depth reduction is impossible for explainable $k$-means and $k$-medians. There has also been some research on other cost functions such as $k$-centers (Laber & Murtinho, 2021) or general $\|\cdot\|_p$ norms (Gamlath et al., 2021). However, to the best of our knowledge, there has not yet been an extension to the closely related kernel $k$-means problem, a gap that we aim to bridge in this work. Other works on interpretable clustering investigate polyhedral descriptions of the clusters (Lawless & Gunluk, 2023), choose similarity-based prototypes for each cluster (Carrizosa et al., 2022), or fit sparse oblique trees to describe the clusters (Gabidolla & Carreira-Perpiñán, 2022).

## B    THE KERNEL K-MEANS ALGORITHM

We first remind the reader of the following result.

**Lemma 1.** *Let $K$ be a kernel operating on data $X \subset \mathbb{R}^d$. Denote $\phi$ for a (possibly data-dependent) feature map of $K$. Let $C_1, \ldots, C_k$ be a partition of $X$ into $k$ clusters with means $c_1, \ldots, c_k \in \mathcal{H}$. Then for any $x \in X$, we have*

$$\arg\min_{l \in [k]} \|\phi(x) - c_l\|^2 = \arg\min_{l \in [k]} \left( \frac{1}{|C_l|^2} \sum_{y,z \in C_l} K(y,z) - \frac{2}{|C_l|} \sum_{y \in C_l} K(x,y) \right) \qquad (8)$$

*Proof.* First recall the identity $\|a - b\|^2 = \|a\|^2 + \|b\|^2 - 2\langle a, b \rangle$ and note that for any cluster $C_l$, its mean is given by

$$c_l = \frac{1}{|C_l|} \sum_{x \in C_l} \phi(x)$$

Expressing everything in terms of the kernel $K$, we see that

$$\|\phi(x) - c_l\|^2 = K(x,x) + \frac{1}{|C_l|^2} \sum_{y,z \in C_l} K(y,z) - \frac{2}{|C_l|} \sum_{y \in C_l} K(x,y)$$

Note that the first term does not depend on $l \in [k]$. $\qquad\qquad \square$

Clearly, equation 8 allows computing distances in the feature space by simply evaluating $K$.

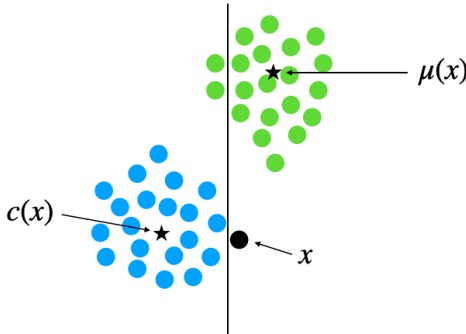

Figure 5: The threshold cut illustrated by the black vertical line defines a decision tree $T$ with 2 leaves. While some points do not end up in the same leaf as their corresponding center, the tree $T$ clearly does a good job in approximating the two clusters.

---

**Algorithm 4** Kernel $k$-means

---

**Input:** Kernel $K$ for data $X \subset \mathbb{R}^d$, integer $k \in \mathbb{N}$
**Output:** Partitioning of $X$ into $k$ clusters $\mathcal{C} = (C_1, \ldots, C_k)$

    Initialize the clusters $C_1, \ldots, C_k$
    Converged $\leftarrow$ FALSE

    **while** Converged $=$ FALSE **do**
        **for** $x \in X$ **do**
            Assign $x$ to $C'_l$ such that $l = \text{argmin}_{j \in [k]} \|\phi(x) - c_j\|^2$ using equation (8)
        **end for**
        **if** $C'_l = C_l$ for all $l \in [k]$ **then**
            Converged $\leftarrow$ TRUE
        **end if**
        Update $\mathcal{C} = (C'_1, \ldots, C'_k)$
    **end while**

---

## C    ITERATIVE MISTAKE MINIMIZATION (IMM) ALGORITHM

In this section, we review the IMM algorithm (Moshkovitz et al., 2020). Consider data $X \subset \mathbb{R}^d$ and centers $\mathcal{M}$ obtained from the $k$-means algorithm. For any $x \in X$, we denote $c(x) \in \mathcal{M}$ for the correct center according to the $k$-means algorithm. The algorithm constructs a decision tree $T$ that sequentially the input space $\mathbb{R}^d$ into $k$ axis-aligned cells using threshold cuts of the form $x_i \gtrless \theta$. At every node $u$ of the tree, IMM chooses the threshold cut that minimizes the number of mistakes. A mistake happens if a point $x$ that arrives at node $u$ together with its correct cluster center $c(x) \in \mathcal{M}$, is now separated from this center as a result of the cut. This happens when $x_i \leq \theta$ but $c_i(x) > \theta$ or vice versa. For any point $x \in X$, we denote $\mu(x)$ for the center that ends up in the same leaf of $T$ as $x$. This is not necessarily $c(x)$, as we illustrate in Figure 5.

Let us also briefly recap the theoretical analysis of IMM for $k$-means, as presented by Moshkovitz et al. (2020). For each node $u$ of the tree $T$ built by IMM, denote by $\mathcal{M}^u$ the remaining set of centers that arrive at the node. The tree $T$ induces a partitioning $\hat{C}_1, \ldots, \hat{C}_k$. Writing $t(u)$ for the number of mistakes at a node $u \in T$ and denoting $D(u) = \max_{a,b \in \mathcal{M}^u} \|a - b\|^2$, it holds that

$$
\begin{aligned}
cost(\hat{C}_1, \ldots, \hat{C}_k) &\leq \sum_{l=1}^{k} \sum_{x \in \hat{C}_l} \|x - \mu(x)\|^2 \\
&\leq \sum_{l=1}^{k} \sum_{x \in \hat{C}_l} \left(2\|x - c(x)\|^2 + 2\|c(x) - \mu(x)\|^2\right) \quad (9) \\
&\leq 2 \cdot cost(C_1, \ldots, C_k) + 2 \sum_{u \in T} t(u)D(u)
\end{aligned}
$$

The main argument in IMM is now given by the following result.

**Lemma 2.** *Let $u \in T$ be any node of the IMM decision tree $T$. Denote $X_{cor}^u$ for the subset of points $x$ that arrive at $u$ together with their correct center $c(x)$.*

*1. For any $i \in [d]$, it holds that*

$$t(u) \cdot \max_{a,b \in \mathcal{M}^u} (a_i - b_i)^2 \leq 4k \cdot \sum_{x \in X_{cor}^u} (x_i - c_i(x))^2 \tag{10}$$

*2. This directly implies*

$$t(u)D(u) \leq 4k \cdot \sum_{x \in X_{cor}^u} \|x_i - c_i(x)\|^2 \tag{11}$$

The proof of the first statement is based on the observation that along every axis $i \in [d]$, and for any threshold cut halfway between two centers $a, b \in \mathcal{M}^u$ (projected to the $i$-th axis), at least $t(u)$ mistakes are made, by definition of $t(u)$. The second statement then directly follows from the first by summing over all $d$ coordinates. Together with Equation 9 and the fact that the maximum depth of the IMM tree is $k$, Lemma 2 yields $O(k^2)$ bounds on the price of explainability. Let us use this opportunity to point out two things.

- Since only threshold cuts halfway between two projected centers are considered in the proof of Lemma 2, the algorithm actually preserves worst-case bounds even when we only check the number of mistakes along these $O(dk)$ threshold cuts, instead of along all possible $O(dn)$ cuts. In other words, the IMM algorithm (as well as our kernelized version) can be run in data-independent time.

- The IMM algorithm is essentially a supervised learning algorithm, recreating a partition $C_1, \ldots, C_k$ no matter how optimal it may be. This observation is crucial when we analyze Kernel IMM on our surrogate features (with respect to a surrogate kernel), for which the reference clustering may not even be close to the optimal partition with respect to this surrogate kernel.

## D  PROOFS FROM SECTION 3

### D.1  PROOF OF THEOREM 1

**Theorem.** *(**The Gaussian kernel cannot have interpretable feature maps**) Consider the Gaussian kernel in $d > 1$ dimensions. There exists a dataset $X$ such that for any feature map $\phi : X \to \mathbb{R}^D$ satisfying $\langle \phi(x), \phi(y) \rangle = K(x, y)$ for all $x, y \in X$, there exists some $j \in [D]$ such that $\phi_j$ depends on more than just one input dimension of $x \in X$.*

*Proof.* We present a simple dataset $X$ such that no feature map for the Gaussian kernel $K(x, y) = \exp(-\gamma \|x - y\|^2)$ can depend only on one input coordinate in each of its elements $\phi_1, \ldots, \phi_D$, no matter how large $D$ may be. Consider three vectors in $\mathbb{R}^2$, given by

$$x^{(1)} = \begin{pmatrix} 0 \\ 0 \end{pmatrix}, x^{(2)} = \begin{pmatrix} 1 \\ 0 \end{pmatrix}, x^{(3)} = \begin{pmatrix} 1 \\ 1 \end{pmatrix}$$

For sufficiently large $\gamma > 0$, we know that $\boldsymbol{K}_{1,2} + \boldsymbol{K}_{2,3} < 1$. Assume there exists some interpretable feature map $\phi : X \to \mathbb{R}^D$. Then, we can write $\phi(x) = (f(x), g(x))$, where $f(x)$ is a function of $x_1$ and $g(x)$ is a function of $x_2$. By our choice of points, we may write

$$\phi(x^{(1)}) = (a, b)$$
$$\phi(x^{(2)}) = (c, b)$$
$$\phi(x^{(3)}) = (c, d)$$

for some vectors $a, b, c, d$. This implies

$$
\begin{aligned}
\boldsymbol{K}_{1,3} &= a^T c + b^T d \\
&= a^T c + \|b\|^2 + \left(b^T d - \|b\|^2\right) \\
&= \boldsymbol{K}_{1,2} + \left(b^T d + \|c\|^2 - 1\right) \\
&= \boldsymbol{K}_{1,2} + \boldsymbol{K}_{2,3} - 1 \\
&< 0
\end{aligned}
$$

a contradiction to $\boldsymbol{K}_{1,3} \geq 0$. For other $\gamma > 0$, simply rescale the data accordingly. $\qquad\square$

## D.2    PROOF OF THEOREM 2

**Theorem.** *Consider the one-dimensional Gaussian kernel $K(x,y) = \exp(-|x-y|^2)$. Then, there exists a dataset $X \subset \mathbb{R}$ such that for any feature map $\phi = (\phi_j)_{j=1}^D$ there exists a component $\phi_j$ that is not monotonic.*

*Proof.* Consider the Gaussian kernel $K(x,y) = \exp\left(-(x-y)^2\right)$ on $\mathbb{R}$. Choose a dataset $X = \{x^{(1)}, x^{(2)}, x^{(3)}\}$ with $x^{(1)} < x^{(2)} < x^{(3)}$ and corresponding kernel matrix $\boldsymbol{K}$ satisfying

$$
\boldsymbol{K}_{1,2} + \boldsymbol{K}_{2,3} < \boldsymbol{K}_{1,3} + 1
$$

This is surely the case when the points are at a sufficiently large distance. Now assume there exists a feature map $\phi : X \to \mathbb{R}^D, \phi(x^{(i)}) = (v_i, w_i)$ where $(v_i)$ is non-decreasing and $(w_i)$ is non-increasing for $i = 1, 2, 3$. In other words, we assume there exist two vectors $v_1, w_1$ as well as non-negative vectors $\epsilon_1, \epsilon_2, \epsilon_3, \delta_1, \delta_2, \delta_3$ such that, writing

$$
\boldsymbol{\Phi} = \left(\begin{array}{ccc} v_1 & v_1 + \epsilon_1 & v_1 + \epsilon_1 + \epsilon_2 \\ w_1 & w_1 - \delta_1 & w_1 - \delta_1 - \delta_2 \end{array}\right) = \left(\begin{array}{ccc} v_1 & v_1 + \epsilon_1 & v_1 + \epsilon_3 \\ w_1 & w_1 - \delta_1 & w_1 - \delta_3 \end{array}\right)
$$

it holds that $\boldsymbol{\Phi}^T \boldsymbol{\Phi} = \boldsymbol{K}$. This condition gives rise to

$$
\begin{aligned}
v_1^T v_1 + w_1^T w_1 &= 1 \\
(v_1 + \epsilon_1)^T (v_1 + \epsilon_1) + (w_1 - \delta_1)^T (w_1 - \delta_1) &= 1 \\
v_1^T (v_1 + \epsilon_1) + w_1^T (w_1 - \delta_1) &= K_{1,2}
\end{aligned}
$$

which implies

$$
v_1^T \epsilon_1 - w_1^T \delta_1 = \boldsymbol{K}_{1,2} - 1
$$

and hence

$$
\|\epsilon_1\|^2 + \|\delta_1\|^2 = 2 - 2\boldsymbol{K}_{1,2}
$$

Using the same arguments for the two other pairs of points from $X$, we derive in the same way

$$
\begin{aligned}
\|\epsilon_2\|^2 + \|\delta_2\|^2 &= 2 - 2\boldsymbol{K}_{2,3} \\
\|\epsilon_3\|^2 + \|\delta_3\|^2 &= 2 - 2\boldsymbol{K}_{1,3}
\end{aligned}
$$

Since $\epsilon_3 = \epsilon_1 + \epsilon_2$ and $\delta_3 = \delta_1 + \delta_2$, this implies

$$
\|\epsilon_1\|^2 + \|\epsilon_2\|^2 + 2\epsilon_1^T \epsilon_2 + \|\delta_1\|^2 + \|\delta_2\|^2 + 2\delta_1^T \delta_2 = 2 - 2\boldsymbol{K}_{1,3}
$$

However, this is equivalent to

$$
\begin{aligned}
2 - 2\boldsymbol{K}_{1,2} + 2 - 2\boldsymbol{K}_{2,3} + 2\left(\epsilon_1^T \epsilon_2 + \delta_1^T \delta_2\right) &= 2 - 2\boldsymbol{K}_{1,3} \iff \\
0 < \epsilon_1^T \epsilon_2 + \delta_1^T \delta_2 = \boldsymbol{K}_{1,2} + \boldsymbol{K}_{2,3} - 1 - \boldsymbol{K}_{1,3} &< 0
\end{aligned}
$$

a contradiction. $\qquad\square$

# E  DECOUPLING DIMENSIONS FOR BOUNDED PRODUCT KERNELS

In this section, we prove that decoupling the input dimensions of bounded product kernels introduces errors no larger than $O(d)$. Recall that our surrogate feature map is given by

$$\phi(x) = (\phi_i(x))_{i=1}^d$$

where each $\phi_i$ is a valid feature map for the one-dimensional kernel $K_i$. This surrogate feature map itself is associated with a surrogate additive kernel:

$$\langle \phi(x), \phi(y) \rangle = \sum_{i=1}^d \langle \phi_i(x), \phi_i(y) \rangle = \sum_{i=1}^d K_i(x, y) \neq K(x, y)$$

Of course, it is not clear that running IMM with respect to this surrogate kernel is actually sensible and not only ensures interpretability, but also preserves worst-case bounds. The justification is given in the following Lemma.

**Lemma 3.** *Let $X \subset \mathbb{R}^d$ be a dataset, partitioned into $k$ clusters $C_1, \ldots, C_k$. Denote by $cost(C_1, \ldots, C_k)$ the kernel $k$-means cost function associated with a distance-based product kernel $K$ on $\mathbb{R}^d$, and by $cost_i$ the cost with respect to the feature map $\phi_i$. Then*

$$cost(C_1, \ldots, C_k) \leq \sum_{i=1}^d cost_i(C_1, \ldots, C_k) \leq d \cdot cost(C_1, \ldots, C_k)$$

*Proof.* First, use the GM-AM inequality and the fact that $K_i(x, y) \leq 1$ to obtain

$$K(x, y) = \prod_{i=1}^d K_i(x, y) \leq \left( \prod_{i=1}^d K_i(x, y) \right)^{1/d} \leq \frac{1}{d} \sum_{i=1}^d K_i(x, y)$$

Thus, rewriting the kernel $k$-means cost function, we see that

$$cost(C_1, \ldots, C_k) = |X| - \sum_{l=1}^k \frac{1}{|C_l|} \sum_{x,y \in C_l} K(x, y)$$

$$\geq |X| - \sum_{l=1}^k \frac{1}{|C_l|} \sum_{x,y \in C_l} \frac{1}{d} \sum_{i=1}^d K_i(x, y)$$

$$= \frac{1}{d} \sum_{i=1}^d \left( |X| - \sum_{l=1}^k \frac{1}{|C_l|} \sum_{x,y \in C_l} K_i(x, y) \right)$$

$$= \frac{1}{d} \sum_{i=1}^d cost_l(C_1, \ldots, C_k)$$

This implies the upper bound from Lemma 3. For the lower bound, note that for any $x, y \in \mathbb{R}^d$

$$\|\phi(x) - \phi(y)\|^2 = 2(1 - K(x, y))$$

$$= 2\left( 1 - \prod_{i=1}^d K_i(x, y) \right)$$

$$\leq 2\left( \sum_{i=1}^d (1 - K_i(x, y)) \right)$$

$$= \sum_{i=1}^d \|\phi_i(x) - \phi_i(y)\|^2$$

where we use the fact that $|\prod_{i=1}^{d} a_i - \prod_{i=1}^{d} b_i| \leq |\sum_{i=1}^{d} a_i - b_i|$ for any $|a_i|, |b_i| \leq 1$. Rewriting the kernel $k$-means cost to make use of this observation, we arrive at

$$
\begin{aligned}
cost(C_1, \ldots, C_k) &= \sum_{l=1}^{k} \frac{1}{2|C_l|} \sum_{x,y \in C_l} \|\phi(x) - \phi(y)\|^2 \\
&\leq \sum_{i=1}^{d} \sum_{l=1}^{k} \frac{1}{2|C_l|} \sum_{x,y \in C_l} \|\phi_i(x) - \phi_i(y)\|^2 \\
&= \sum_{i=1}^{d} cost_i(C_1, \ldots, C_k)
\end{aligned}
$$

$\square$

# F INTERPRETABLE TAYLOR KERNELS

## F.1 PROOF OF THEOREM 3

**Theorem.** *(**Threshold cuts in the surrogate feature space of interpretable Taylor kernels yield interpretable decision trees**) Let $K$ be product of interpretable Taylor kernels, and let $\phi$ denote its surrogate feature map. Then, a threshold cut of the form $\phi_{i,j}(x_i) \gtrless \theta$ leads to an interpretable decision tree in $\mathbb{R}^d$.*

*Proof.* Recall our notion of surrogate features for interpretable Taylor kernels: For all $i \in [d]$ and $x \in X$, we define

$$
\phi_i(x) = \left( f(x_i) x_i^j \sqrt{\frac{g^{(j)}(0)}{j!}} \right)_{j=0}^{M}
$$

and refer to the concatenation $\phi(x) = (\phi_i(x))_{i=1}^{d}$ as the surrogate feature map of order $M$. Note that $\phi'_{i,j}(x) \propto x_i^{j-1} (f'(x_i) x_i + j f(x_i))$. For interpretable Taylor kernels, this expression is zero for at most one point $x$ in the domain of the kernel. Thus, when we run IMM on these surrogate features, the threshold cuts $\phi_{i,j}(x) \gtrless \theta$ can be translated to interpretable decision trees in the input space. $\square$

## F.2 PROOF OF THEOREM 4

**Theorem.** *Let $C_1, \ldots, C_k$ be the clusters of a dataset $X \subset \mathbb{R}^d$ derived from kernel $k$-means, where the kernel $K$ is an interpretable Taylor kernel. Denote by $\mathbf{\Phi} = \phi(X)$ the surrogate features for $K$ on $X$. Then, the interpretable decision tree obtained from Kernel IMM on $\mathbf{\Phi}$ satisfies $p(T, X) = O\left(dk^2\right) + \frac{O(dk^2\delta)}{cost_{opt}(X)}$, where $\delta = \delta(M)$ depends on the order $M$ of the surrogate feature map and $\lim_{M \to \infty} \delta(M) = 0$ for any dataset $X$. Thus, by adaptively choosing $M$ such that $\delta = O(cost_{opt})$, we obtain $O(dk^2)$ bounds.*

*Proof.* We begin by noting that for all $i \in [d]$ and $x, y \in X$, and any integer $M$, Taylor's formula ensures that

$$
\begin{aligned}
|K_i(x,y) - \langle \phi_i(x), \phi_i(y) \rangle| &= f(x_i) f(y_i) \cdot \left| \sum_{j=M}^{\infty} \frac{g^{(j)}(0)}{j!} (x_i y_i)^j \right| \\
&\leq \left( \max_{x \in X} |f(x_i)|^2 \right) \cdot \frac{\|g^{(j)}\|_{\infty([0, x_i y_i])} (x_i y_i)^M}{M!}
\end{aligned}
$$

Since $g$ is assumed to be analytic, $\lim_{M \to \infty} \langle \phi_i(x), \phi_i(y) \rangle = K_i(x,y)$ for all $x, y$. Thus, the surrogate features $\phi_i$ approximate the one-dimensional kernels $K_i$ within some $\delta > 0$ that vanishes as $M \to \infty$. Now, fix a dataset $X$ and let $\hat{C}_1, \ldots \hat{C}_k$ be the interpretable clusters chosen by IMM

on the surrogate features $\phi$. Denote by $C_1, \ldots, C_k$ the reference clusters obtained from unrestricted kernel $k$-means with respect to the original, interpretable Taylor kernel $K$. Denote by $cost$ the kernel $k$-means cost with respect to the product kernel $K$, by $cost_i$ the costs with respect to the univariate kernels $K_i$, and by $\widetilde{cost_i}$ the cost with respect to the approximate kernel implicitly defined by virtue of the surrogate feature maps $\phi_i$. Using Lemma 3 from Appendix E and keeping track of the approximation error, we obtain

$$
\begin{aligned}
cost(\hat{C}_1, \ldots, \hat{C}_k) &\leq \sum_{i=1}^{d} cost_i(\hat{C}_1, \ldots, \hat{C}_k) \\
&\leq \sum_{i=1}^{d} \left( \widetilde{cost_i}(\hat{C}_1, \ldots, \hat{C}_k) + O(\delta) \right) \\
&\leq O(d\delta) + \sum_{i=1}^{d} \left( O(k^2) \cdot \widetilde{cost_i}(C_1, \ldots, C_k) \right) \\
&\leq O(d\delta) + \sum_{i=1}^{d} O(k^2) \left( cost_i(C_1, \ldots, C_k) + O(\delta) \right) \\
&= O(dk^2\delta) + \sum_{i=1}^{d} O(k^2) \cdot cost_i(C_1, \ldots, C_k) \\
&\leq O(dk^2) \cdot cost(C_1, \ldots, C_k) + O(dk^2\delta)
\end{aligned}
$$

$\square$

# G  DISTANCE-BASED PRODUCT KERNELS

Consider a distance-based product kernel $K(x, y) = \prod_{i=1}^{d} h(|x_i - y_i|)$ on $\mathbb{R}^d$. As pointed out in the main paper, we may define surrogate features via $\phi_{i,j}(x) = K_i(x, x^{(j)})$ were $x^{(j)}$ denotes the $j$th point in the dataset $X$. While these features lead to interpretable decision trees, they do not define features that approximate the one-dimensional kernel: For all $i \in [d]$, it must be noted that $\langle \phi_i(x), \phi_i(y) \rangle \neq K_i(x, y)$ under the standard Euclidean inner product. Of course, if we equip $\mathbb{R}^n$ with a new inner product given by $\langle \phi_i(x), \phi_i(y) \rangle = \phi_i(x)^T \boldsymbol{K}_i^{-1} \phi_i(y)$, then our surrogate features remain valid features for the one-dimensional kernels $K_i$. Consequently, when setting $\phi = (\phi_i)_{i=1}^{d}$, we actually map to a space $\mathcal{V}$ in which the inner product is given by $\langle u, v \rangle = \sum_{i=1}^{d} u_i^T \boldsymbol{K}_i^{-1} v_i$. Running explainable clustering algorithms such as IMM in this space is possible, it however introduces additional constants that depend on the kernel matrices $\boldsymbol{K}_i$.

**Theorem.** *Let $C_1, \ldots, C_k$ be the clusters of a dataset $X \subset \mathbb{R}^d$ derived from kernel $k$-means, where the kernel $K$ is a distance-based product kernel. Denote by $\boldsymbol{\Phi} = \phi(X)$ the surrogate features for $K$ on $X$, where $\phi_{i,j}(x) = K_i(x, x^{(j)})$. Then, the interpretable decision tree obtained from Kernel IMM on $\boldsymbol{\Phi}$ satisfies $p(T, X) = O\left(Cdk^2\right)$, where $C$ depends on the dataset $X$ and the kernel $K$.*

*Proof.* To prove Theorem 5 we adopt the general proof strategy from IMM and again decouple dimensions as discussed in Appendix E. Again, $cost_i$ denotes the cost with respect to each one-dimensional kernel $K_i$.

$$
\begin{aligned}
cost(\hat{C}_1, \ldots, \hat{C}_k) &\leq \sum_{i=1}^{d} cost_i(\hat{C}_1, \ldots, \hat{C}_k) \\
&= \sum_{l=1}^{k} \sum_{x \in \hat{C}_l} \| \phi(x) - \mu(x) \|_{\mathcal{V}}^2 \quad\quad (12) \\
&\leq 2 \cdot cost_{\mathcal{V}}(C_1, \ldots, C_k) + 2 \sum_{u \in T} t(u) D(u)
\end{aligned}
$$

Now, we would in principle like to use Lemma 2. However, the space $\mathcal{V}$ in which Kernel IMM runs is equipped with the non-Euclidean inner product. Thus, the second part (11) of Lemma 2 no longer follows from the first (10). Instead, we arrive at

$$
\begin{aligned}
t(u)D(u) := t(u) \cdot & \max_{a,b \in \mathcal{M}^u} \|a - b\|_{\mathcal{V}}^2 \\
= t(u) \cdot & \max_{a,b \in \mathcal{M}^u} \sum_{i=1}^{d} (a_i - b_i)^T \boldsymbol{K}_i^{-1} (a_i - b_i) \\
\leq \kappa_1 \cdot t(u) \cdot & \max_{a,b \in \mathcal{M}^u} (a - b)^T (a - b) \\
\leq 4\kappa_1 \cdot k & \sum_{x \in X_{cor}^u} (\phi(x) - c(x))^T (\phi(x) - c(x)) \\
\leq 4\kappa_1 \kappa_2 \cdot k & \sum_{x \in X_{cor}^u} \|\phi(x) - c(x)\|_{\mathcal{V}}^2
\end{aligned}
$$

where we use the first part of Lemma 2 in the second inequality. The constants $\kappa_1, \kappa_2$ are given by

$$
\begin{aligned}
\kappa_1 &= \max_{a,b \in \mathcal{M}^u} \frac{\sum_{i=1}^{d} (a_i - b_i)^T \boldsymbol{K}_i^{-1} (a_i - b_i)}{\sum_{i=1}^{d} (a_i - b_i)^T (a_i - b_i)} \\
\kappa_2 &= \max_{x \in X} \frac{\sum_{i=1}^{d} (\phi_i(x) - c_i(x))^T (\phi_i(x) - c_i(x))}{\sum_{i=1}^{d} (\phi_i(x) - c_i(x))^T \boldsymbol{K}_i^{-1} (\phi_i(x) - c_i(x))}
\end{aligned}
$$

and depend on the data. $\qquad \square$

## H  KERNEL IMM FOR ADDITIVE KERNELS

Additive kernels are of the form $K(x,y) = \sum_{i=1}^{d} K_i(x,y)$ where every individual $K_i$ is a kernel on a suitable subset of $\mathbb{R}$, often the positive real line. Feature maps of additive kernels $K$ can be written as the concatenation of the univariate feature maps associated with each kernel $K_i$. If these univariate — and hence interpretable — maps consist of functions that have at most one change of slope, threshold cuts with respect to the feature map lead directly to interpretable decision trees. Conveniently, this is the case for several important additive kernels.

- Consider Hellinger's kernel $K(x,y) = \sum_{i=1}^{d} \sqrt{x_i y_i}$ on a dataset $X \subset [0, \infty)^d$. Then $K$ admits the simple interpretable (and monotonic) feature map $\phi(x) = (\sqrt{x_1}, \dots, \sqrt{x_d})$.

- Let $K(x,y) = \sum_{i=1}^{d} \min\left(x_i^\beta, y_i^\beta\right)$ be the generalized histogram intersection kernel, for some parameter $\beta > 0$. For any dataset $X \subset [0, \infty)^d$, there again exists a feature map that yields interpretable decision trees: Given a feature $i \in [d]$, denote by $z_1 < \cdots < z_{m_i}$ the $m_i$ unique values in the set $\{x_i^\beta \mid x \in X\}$. For every component $j \in [m_i]$, let

$$
\phi_{i,j}(x) = \begin{cases} \sqrt{z_j - z_{j-1}} & \text{, if } j \geq 2 \text{ and } x_i^\beta \geq z_j \\ \sqrt{z_j} & \text{, if } j = 1 \\ 0 & \text{, else} \end{cases} \tag{13}
$$

  Then the concatenation $\phi = (\phi_{i,j})_{i \in [d], j \in [m_i]}$ is composed of monotone functions. Moreover, it is straightforward to verify that for all $x, y \in X$ it holds that $K(x,y) = \langle \phi(x), \phi(y) \rangle$. Thus, $\phi$ is a valid feature map.

- As a third example, consider the additive $\chi^2$ kernel $K(x, y) = \sum_{i=1}^{d} \frac{2x_i y_i}{x_i + y_i}$ on a dataset $X \subset (0, \infty)^d$. Given some $M \in \mathbb{N}$, the kernel $K$ can be approximated from

$$
\begin{aligned}
K(x, y) &= \sum_{i=1}^{d} 2x_i y_i \cdot \int_0^1 t^{x_i + y_i - 1} dt \\
&= \sum_{i=1}^{d} \int_0^1 \sqrt{2/t} \cdot x_i t^{x_i} \cdot \sqrt{2/t} \cdot y_i t^{y_i} dt \qquad (14) \\
&\approx \sum_{i=1}^{d} \frac{1}{M} \sum_{j=1}^{M} \sqrt{2M/j} \cdot x_i \left(\frac{j}{M}\right)^{x_i} \cdot \sqrt{2M/j} \cdot y_i \left(\frac{j}{M}\right)^{y_i}
\end{aligned}
$$

We conclude that the concatenation of all functions

$$
\phi_{i,j}(x) = \sqrt{\frac{2}{j}} \cdot x_i \left(\frac{j}{M}\right)^{x_i}
$$

defines an approximate feature map for $K$. Since each $\phi_{i,j}$ is strictly increasing for $x_i < 1/\log(M/j)$, and strictly decreasing for $x_i > 1/\log(M/j)$, they give rise to a feature map for which threshold cuts lead to interpretable decision trees. For normalized histograms we find that even low values of $M \leq 5$ induce the same partitions as the additive $\chi^2$ kernel, and hence $O(k^2)$ bounds are preserved when running (Kernel) IMM with respect to the kernel induced by the map $\phi$.

# I EXPERIMENTS

Let us now give some details on the experiments. Our code is available on GitHub.

**Main experiments for Kernel IMM.** We first verify the approximation properties of our proposed methods on the synthetic datasets "Pathbased", "Aggregation" and "Flame" (Fränti & Sieranoja) which have $k = 3$, $k = 7$ and $k = 2$ clusters respectively. We start with linear $k$-means and IMM on all three, and then run kernel $k$-means with both the Laplace as well as the Gaussian kernel over a range of hyperparameters $\gamma$, choosing the best agreement with the ground truth as our starting point for Kernel IMM. When the Gaussian kernel is chosen, we run Kernel IMM both on the surrogate Taylor features from Definition 3 with $M = 5$, as well as on the surrogate features based on the kernel matrix, as defined in Equation (4), and choose the better one. We then refine the partition induced by Kernel IMM using both Kernel ExKMC as well as Kernel Expand, constructing $m = 6$, $m = 10$ and $m = 4$ leaves respectively. Note that at every step, Kernel ExKMC and Kernel Expand only need to check the cost of the threshold cuts at the new nodes (obtained from the previous iteration). Thus, adding $m$ leaves to an existing tree with $p$ leaves amounts to $p + 2m$ iterations over all possible threshold cuts.

We follow the same procedure for the two real world datasets. In **Iris** (Fisher, 1936), there are three classes with 50 observations each. Every class refers to a type of iris plant. As illustrated in the barplot included in Section 6, kernel $k$-means slightly improves over $k$-means and this translates to Kernel IMM, Kernel ExKMC and Kernel Expand. The **Wisconsin breast cancer** dataset consists of 569 observations of benign and malignant cells. The 30-dimensional features describe characteristics of the cell nuclei observed in each image. Interestingly, IMM exactly replicates its suboptimal reference $k$-means clustering. Kernel $k$-means better identifies the ground truth, and Kernel IMM approximates it well (even achieving a slightly higher agreement with the ground truth). The same is true for Kernel ExKMC and Kernel Expand.

**Kernel IMM for the $\chi^2$ kernel.** The additive $\chi^2$ kernel is evaluated on a toy dataset obtained from a mixture model of four discrete distributions, with values in four bins. Figure 6 shows a plot of the different distributions. For all four distribution, we draw 5 instances of 100 random samples, and compute the fraction of observations in each bin for every instance (thus $n = 20$ and $d = 4$). We repeat this procedure 100 times. We find that the $\chi^2$ kernel achieves a Rand index that is consistently higher than the one of $k$-means (see Figure 6). This is not very surprising: The denominator of the

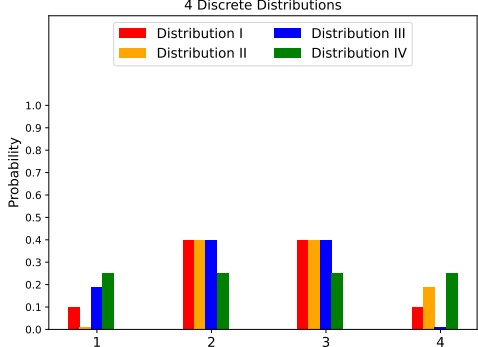
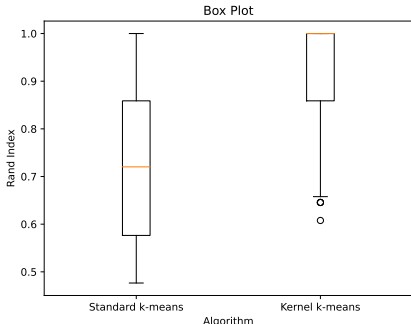

Figure 6: We cluster samples drawn from a mixture model of four discrete distributions by checking the fraction of observations in each of four bins. The true underlying probabilities are shown in the left plot. Over 100 draws of samples, the $\chi^2$ kernel improves over standard $k$-means in recovering the ground truth, as the boxplot on the right shows.

$\chi^2$ kernel accounts for the overall number of observations in each bin, penalizing deviations in less probable bins more than in frequently visited bins — a nonlinear characteristic that standard $k$-means lacks.

To provide some more intuition on how Kernel IMM constructs interpretable decision trees, let us now give some additional details for the $\chi^2$ kernel. Having drawn 5 instances (containing 100 samples) for each distribution, we compute the class probabilities for the four bins and cluster using kernel $k$-means. We then map to a higher-dimensional feature space using the features derived for the $\chi^2$ kernel (see Appendix H). We choose $M = 5$ features for every dimension, and hence operate in a $D = Md = 20$-dimensional space. Assuming the first threshold cut is chosen along the first axis in the feature space, Kernel IMM provides us with some $\theta$ such that we cut at

$$\phi_{1,1}(x) \leq \theta \iff \sqrt{2}x_1 \left(\frac{1}{5}\right)^{x_1} \leq \theta \iff \frac{x_1}{5^{x_1}} \leq \frac{\theta}{\sqrt{2}}$$

All we are now left with is identifying which values of $x_1$ satisfy the above inequality, and which do not. Equivalently, we can represent this as an interval (or its complement) in the sample space.

**Comparison with CART.**   Standard decision tree algorithms such as CART also perform well in our experiments. CART achieves a price of explainability very close to the one that Kernel IMM attains (despite it being known that no approximation results exist for CART in the standard $k$-means setting). We show a comparison of the results in Table 2. For the Wisconsin breast cancer dataset, CART even improves over unconstrained kernel $k$-means (which has, in this case, not found the optimal partition).

Table 2: Comparison of price of explainability (PoE) between CART with one-sided cuts and $k$ leaves, and Kernel IMM.

| Dataset | Kernel | PoE (Kernel IMM) | PoE (CART) |
|---|---|---|---|
| Pathbased | Gaussian | 1.06645 | 1.07004 |
| Aggregation | Laplace | 1.00125 | 1.00125 |
| Flame | Gaussian | 1.02256 | 1.02732 |
| Iris | Laplace | 1.00502 | 1.00502 |
| Cancer | Gaussian | 1.00179 | 0.99330 |

## J    EXKMC ON AN EMPTY TREE ADMITS NO WORST-CASE GUARANTEES (THEOREM 6)

**Theorem.** *(ExKMC admits no worst-case bounds) For any $m \in \mathbb{N}$ there exists a data set $X \subset \mathbb{R}$ such that the price of explainability on $X$ is $p(X) = 1$, IMM attain this minimum, but ExKMC (initialized on an empty tree) constructs a decision tree $T$ with $p(T, X) \geq m$.*

*Proof.* Consider a dataset $X$ of size $n$ in $\mathbb{R}$, consisting of three clusters with centers given by $c_1 = -1$, $c_2 = 0$, $c_3 = 1$. Assume all points of each cluster $C_i$ lie at a distance of exactly $\epsilon$ from the center $c_i$, half of them at $c_i - \epsilon$ and half of them at $c_i + \epsilon$. The optimal $k$-means cost is then given by

$$cost_{opt} = n\epsilon^2$$

At the first iteration of ExKMC, suppose it chooses some cut that does not separate points belonging to the same cluster. We may then assume that w.lo.g. $(i, \theta) = (1, 0.5)$, which implies $c_R = c_3$ and w.l.o.g. $c_L = c_1$. Thus, the total cost of this cut is

$$f(i, \theta) = \frac{2n}{3}\epsilon^2 + \sum_{x \in C_2} \|x - c_1\|^2$$
$$= \frac{2n}{3}\epsilon^2 + \frac{n}{6}(1 - \epsilon)^2 + \frac{n}{6}(1 + \epsilon)^2$$

This cost can be improved upon when ExKMC cuts along $(i, \theta) = (1, 0)$, choosing $c_1 = c_L$, $c_3 = c_R$, leading to a cost of

$$f(i, \theta) = \frac{2n}{3}\epsilon^2 + \frac{n}{3}(1 - \epsilon)^2$$

Since ExKMC can now only add one more leaf, it inevitably separates at least $\frac{n}{6}$ points from their correct cluster center. Thus, there exists a leaf for which $\frac{2n}{6} \cdot \frac{n}{3} = \frac{n^2}{9}$ pairs of points are at distance of at least $1 - 2\epsilon$, and the final cost of the associated decision tree $T$ is hence

$$cost(T) \geq \frac{n\epsilon^2}{2} + \frac{1}{n} \cdot \frac{2n^2(1 - 2\epsilon)^2}{9} = \frac{n\epsilon^2}{2} + \frac{2n(1 - 2\epsilon)^2}{9}$$

The price of explainability of $T$ is given by

$$p(T, X) \geq \frac{n\epsilon^2}{2n\epsilon^2} + \frac{2(1 - 2\epsilon)^2}{9\epsilon^2}$$

Thus, as $\epsilon \to 0$, we see that ExKMC gives rise to decision trees with an arbitrarily large price of explainability $p(T, X) \geq m$. Note that IMM recreates the optimal partition perfectly, since it is possible to split clusters without making any mistakes. Thus $p(X) = 1$. Clearly, the fact that ExKMC restricts our choice of centers to the set of reference centers limits its performance.    □

## K    THE PRICE OF EXPLAINABILITY CAN BE INFINITE FOR KERNEL CLUSTERING (THEOREM 7)

The price of explainability, as defined in Definition 4, may not be finite for all kernels.

**Proposition 1.** *(Unbounded price for the quadratic kernel) Let $K(x, y) = \langle x, y \rangle^2$. Then, there exists a dataset $X \subset \mathbb{R}^2$ such that $p(X) = \infty$.*

*Proof.* Let $x = (0, 1), y = (0, -1), z = (1, 0), w = (-1, 0) \in \mathbb{R}^2$. Consider the quadratic kernel $K(s, t) = (\langle s, t \rangle)^2$ operating on $X = \{x, y, z, w\}$. Then quadratic kernel 2-means achieves a cost of zero by partitioning $X$ into the two clusters $C_1 = \{x, y\}, C_2 = \{z, w\}$, since $K(s, t) = 1$ for all $s, t \in X$ from the same cluster. However, no interpretable decision tree can reproduce this partitioning of $X$. Every decision tree $T$ with two leaves assigns one pair of points $(s, t)$ from different clusters to the same leaf. Since $K(s, t) = 0$ for this pair, $cost(T) > 0$ and thus $p(X) = \infty$.    □

This may not come as a surprise considering the special geometry of the quadratic kernel. However, the following result demonstrates that even distance-based kernels can lead to similar problems. For this, we consider the $\epsilon$-neighborhood kernel $K(x, y) = \mathbf{1}(\|x - y\| < \epsilon)$. While $K$ is **not** a positive definite kernel (its kernel matrix may have eigenvalues), it is commonly used in spectral and kernel clustering nonetheless.

**Proposition 2.** *(Unbounded price for the $\epsilon$-neighborhood kernel) Let $\epsilon > 0$ and let $K(x, y) = \mathbf{1}(\|x - y\| \leq \epsilon)$. Then, there exists a dataset $X$ such that $p(X) = \infty$.*

*Proof.* For four points $x^{(1)}, \ldots, x^{(4)}$, there are exactly $d' = 6$ ways of assigning to two of them the value 0, and to the two others the value 1. Similarly, for four other points $x^{(5)}, \ldots, x^{(8)}$, there are exactly $d' = 6$ ways of assigning to two of them the value 0, and to the two others the value $-1$. There are $d = (d')^2 = 36$ possible combinations of all these assignments. We define a $d$-dimensional cluster $C_1$ by concatenating all of the combinations for each of the points $x^{(1)} \ldots, x^{(4)}$. Thus, every $x \in C_1$ is a vector in $\{0, 1\}^d$. Similarly, define $C_2$ by concatenating all of the combinations for each of the points $x^{(5)}, \ldots, x^{(8)}$. Thus, the vectors $y \in C_2$ are $\in \{0, -1\}^d$. Note that for any $x, x' \in C_1$, they agree in exactly $d/3$ dimensions but differ by 1 everywhere else. Thus $\|x - x'\|^2 = 24$. The same holds for any pair $y, y' \in C_2$. However, for any pair $x \in C_1, y \in C_2$ from different clusters, they agree on $d/4 = 9$ dimensions, but are separated by 1 along $d/2 = 18$ dimensions and are separated by 2 elsewhere. Thus, their distance is $\|x - y\|^2 = 18 + 36 = 54$. By choosing the $\epsilon$-neighborhood kernel with $\epsilon = \sqrt{24}$, we see that $K(x, x') = K(y, y') = 1$ for all $x, x' \in C_1, y, y' \in C_2$. This implies an optimal kernel $k$-means cost of zero. However, any threshold cut along any one of the $d$ dimensions will assign at least one pair $(x, y)$ from distinct clusters to the same leaf, implying $K(x, y) = 0$. Thus, the kernel $k$-means cost for this explainable partition is strictly positive. This proves $p(X) = \infty$. To produce the same result for other values of $\epsilon > 0$, simply rescale the data accordingly.

$\square$

