# OpenReview forum: "Explaining Kernel Clustering via Decision Trees"
_ICLR.cc/2024/Conference — ICLR 2024 poster_

### Official Review · Reviewer_aXrD · 2023-10-29

**Soundness:** 4 excellent
**Presentation:** 4 excellent
**Contribution:** 3 good
**Rating:** 8
**Confidence:** 5

**Summary:**

This paper proposes an explainable kernel clustering algorithm. They consider the explainable clustering that uses a threshold decision tree to partition the data set. Their algorithm first computes a kernel k-means clustering on the data set. Then, the algorithm converts the kernel clustering into a (generalized) threshold decision tree. Each internal node of the threshold decision tree given by the algorithm is a threshold cut (or an interval cut with two thresholds) on a single feature of the original feature space.
They also prove the upper bound on the worst-case price of explainability for kernel k-means clustering. They define the price of explainability for kernel k-means clustering as the ratio of the cost given by the decision tree and the cost of optimal kernel k-means clustering. They show that their kernel IMM algorithm achieves O(k^2) upper bound for additive kernels, O(dk^2) upper bound for interpretable Taylor kernels, and O(Cdk^2) for distance-based product kernels, where C depends on the dataset X and the kernel K.

**Strengths:**

1.	The paper is well-written and easy to follow.

2.	The paper proposed a new problem explainable kernel clustering and an algorithm for it. This problem generalized the explainable clustering problem proposed by Dasgupta et al (2020). The explainable clustering problem has been extensively studied in recent years. The explainable k-means and k-medians clustering generate a threshold decision tree clustering which is easy to understand and visualized by humans. In practice, the data set might not be well-clustered by k-means and k-medians clustering in the original feature space. The kernel method is widely used to map the original data into a well-clustered space. The algorithm utilizes kernel clustering to create an explainable clustering in the original feature space. The problem is well-motivated and interesting.

3.	They also provide theoretical analysis on the price of explainability for kernel k-means clustering. For several popular kernels, they show interesting upper bounds on the worst-case price of explainability.

**Weaknesses:**

1. Their kernel IMM algorithm is a direct generalization of the IMM algorithm proposed by Dasgupta et al (2020) to the kernel clustering. Although they compare the cost of the explainable clustering to the cost of optimal kernel k-means clustering, their upper bound loses a factor of d for interpretable Taylor kernels and distance-based product kernels. The dimension d might be very large in real-world data sets.

2. They consider the generalized threshold decision tree, in which each internal node can partition the space by an interval on a single feature. The interval cuts can be seen as two threshold cuts. Therefore, this generalized threshold decision tree can be converted to a regular threshold decision tree with more than k leaves. The previous works showed that expanding the threshold decision tree to more than k leaves can reduce the clustering cost. Thus it is unclear to me whether the improvement in the clustering cost in the experiments are due to the better kernel clustering or due to these generalized decision tree structure.

**Questions:**

1 Would this generalized threshold decision tree (with interval cut) significantly reduce the clustering cost since it partitions the space into more parts? It would be interesting to evaluate the effect of this tree structure change by converting it back to a regular threshold tree and comparing it with the expanded IMM with the same number of leaves.

---

> ### Author Response · Authors · 2023-11-18
> **Author response**
>
> We thank you for your review. We are glad that you enjoyed the paper, and we appreciate that you agree that the problem is both interesting and relevant. We now address the weaknesses mentioned in your review.
>
> **The dimension $d$ might be very large in real-world data sets.**
>
> We agree that the upper bounds can be loose for well-clustered data. This is not very surprising. Indeed, previous works on explainable $k$-means have also reported approximation ratios of close to $1$ in practice (for example, see Table 1 in [1]). In our case, the additional factor of order $d$ is a consequence of decoupling dimensions in the kernel function, which is necessary to obtain interpretable feature maps. It is likely that algorithms tailored to specific kernels allow for tighter bounds (much in the same way as for $k$-means, $k$-medians and higher $\|\cdot \|_p$-norms, distinct algorithms have emerged [2]). Moreover, better guarantees could possibly be given under assumptions on the underlying distribution. These are very interesting avenues to explore in the future. However, since explainable kernel clustering has not been studied prior to our work, this paper takes a broader perspective, proposing general methods that work for a large class of kernels, regardless of the underlying distribution.
>
> [1] Laber, Eduardo S., and Lucas Murtinho. "On the price of explainability for some clustering problems." International Conference on Machine Learning. PMLR, 2021.
>
> [2] Gamlath, Buddhima, et al. "Nearly-tight and oblivious algorithms for explainable clustering." Advances in Neural Information Processing Systems 34 (2021): 28929-28939.
>
> **It is unclear to me whether the improvement in the clustering cost in the experiments are due to the better kernel clustering or due to these generalized decision tree structure. Would this generalized threshold decision tree (with interval cut) significantly reduce the clustering cost since it partitions the space into more parts? It would be interesting to evaluate the effect of this tree structure change by converting it back to a regular threshold tree and comparing it with the expanded IMM with the same number of leaves.**
>
> While it is true that the proposed interval cuts allow for more refined partitions of the input space, let us briefly emphasize that they do this in a way that arguably does not harm interpretability at all (unlike a decision tree with twice the number of nodes). From a theoretical point of view, interval cuts are justified by Theorem 2. The improved performance of our methods (over those from explainable $k$-means) is primarily a consequence of us approximating partitions that more accurately reflect meaningful cluster structures. Unless the data exhibits linear cluster structures, standard explainable $k$-means is unable to identify the correct partitions, since it relies on the centers that $k$-means provides us with, and hence typically would inherit its deficiencies. Regarding the question raised by the reviewer: Kernel IMM (after unfolding it back to a tree with twice the number of nodes) is more restricted in its structure than Kernel Expand (which could, for example, access twice the number of distinct feature dimensions in all of its nodes).

---

> > ### Comment · Reviewer_aXrD · 2023-11-19
> >
> > Thank the authors for the detailed response. I would like to keep my score.
> >
> > I agree that this interval-cut tree may not harm the interpretability too much since kernel IMM uses only k-1 features. While, previous work like ExKMC would choose more than k-1 features.
> >
> > I just would like to clarify about my question. In experiments, like Figure 4, I think these methods have different model complexity (interpretability), (IMM>Kernel IMM > Kernel ExKMC, Kernel Expand) because IMM only uses threshold cuts on k-1 features, Kernel IMM uses interval cuts on k-1 features. My question is about comparing Kernel IMM with non-Kernel ExKMC (both have the same number of leaves in the threshold cut tree structure.). I thought it would be an interesting comparison because in terms of model complexity (interpretability) Kernel IMM is even better than non-Kernel ExKMC.
> > I just think it might be an interesting experiment to try after the response.

---

### Official Review · Reviewer_762t · 2023-10-30

**Soundness:** 3 good
**Presentation:** 3 good
**Contribution:** 3 good
**Rating:** 6
**Confidence:** 3

**Summary:**

This work aims to fill the gap in the literature on (inherently) interpretable clustering methods. Specifically, this work proposes algorithms to approximate kernel k-means methods. Similar approaches have been proposed before for classic k-means algorithms, however practice oftentimes requires more flexible clustering methods such as kernel k-means. The authors propose a variant of the interpretable clustering method Iterative Mistake Minimization (IMM), which is an algorithm that approximates a given k-means clustering by a decision tree with k leaves, where each leaf represents a cluster. The proposed algorithm (Kernel IMM) builds on IMM, and essentially constructs a decision tree to approximate the clusters induced by kernel k-means. Besides Kernel IMM they also present Kernel ExKMC and Kernel Expand, two methods where new leaves are added in a greedy approach to improve accuracy of the tree (potentially at a loss of interpretability). Performance of the approaches is evaluated on a few datasets.

**Strengths:**

The paper is well written and has a clear structure. The motivation of the proposed algorithm(s) is clear and the paper addresses a relevant and timely issue. The paper discusses central ideas in a concise manner, with elaborations in the appendix. The approach seems novel in a sense that it extends ideas from interpretable k-means to kernel k-means, which has not been done before. The chosen figures support the manuscript, and especially Figure 2 is a nice visualisation of the proposed method.

**Weaknesses:**

see the main weakness of this paper in the limited experimental results. It would be nice to see a bit more elaborate experiments. The authors claim that the resulting trees are interpretable, as decision trees are generally understood to be both globally and locally interpretable. However, even decision trees may become uninterpretable with too many leaves and/or too deep paths. Hence, it would be interesting to see the size of the trees. In Figure 4 (right) it can be seen that Kernel IMM performs (much) worse than kernel k-means, while Kernel ExKMC and Kernel Expand show a good performance compared to kernel k-means — however at what price of interpretability? To me, it would be interesting to see a bit more elaboration and analysis of the interpretability (and the trade-off). Presenting a sort of a case study on one of the datasets may be a possible option to show such results.

Minor comments:
- why does (the reference to) appendix C appear before appendix B in the manuscript? They could be switched
- the introduction may be shortened (especially 1st and 2nd paragraph) to allow for more space to report experimental results
- it would be nice to include some references to works on (inherently) interpretable clustering methods that do not necessarily build on k-means (for example Carrizosa et al. (2023) or works on cluster description (e.g. Lawless & Günlük (2023))
- contributions could be listed for conciseness
- 2x “of” in paragraph on explainable k-means

**Questions:**

Is it possible to compare to other works constructing interpretable models (not necessarily based on k-means or kernel k-means), for example in terms of accuracy or interpretability (nr. of rules, for example)? If not, how so?

---

> ### Author Response · Authors · 2023-11-18
> **Author response**
>
> We thank you for your review. We appreciate that you find the paper to be well-written and the problem to be novel and relevant. We will incorporate the additional references and minor comments in a revision. We address the weaknesses mentioned in the review below.
>
> **It would be nice to see a bit more elaborate experiments. Is it possible to compare to other works constructing interpretable models (not necessarily based on k-means or kernel k-means), for example in terms of accuracy or interpretability (nr. of rules, for example)? If not, how so?**
>
> A systematic empirical evaluation of different tree-based clustering algorithms is interesting, but would take considerable time and resources, and will not be possible within the timeline for these discussions. We will consider this in the future, and note that existing papers from this line of research (published at similar ML conferences) also restrict to a small set of experiments on standard clustering datasets.
>
> Regarding the complexity of the decision trees: Kernel IMM always fits a tree with exactly $k$ leaves. On the synthetic clustering datasets (containing $k=3$, $k=7$ and $k=2$ clusters respectively), we refine the partitions of Kernel IMM using Kernel ExKMC and Kernel Expand, fitting $m = 6$, $m = 10$ and $m = 4$ leaves respectively. The price of explainability is included in Figure 4 (left plot). For the *pathbased* dataset, we have illustrated the suboptimal performance of Kernel IMM in Figure 3, as it was one of the primary reasons for us to explore the greedy algorithms derived in Section 5.

---

> ### Comment · Reviewer_762t · 2023-11-22
> **Reply to authors' response**
>
> Thank you for the response. I read it. I keep my score as it is.

---

### Official Review · Reviewer_8Bws · 2023-10-31

**Soundness:** 3 good
**Presentation:** 3 good
**Contribution:** 2 fair
**Rating:** 8
**Confidence:** 4

**Summary:**

This manuscript extends the study of explainable $k$-means to the kernel $k$-means, also using decision trees as the "representation" of explanation but the node does not characterize axis-parallel threshold cuts anymore. In fact, the explainability is relaxed to be an interval or the complement of it. To address the issue of non-injectivity of cluster centers from the RKHS to $\mathbb{R}^d$, the decision is also made on the proposed surrogate feature maps instead of Euclidean space. Based on these new notions and the previous Iterative Mistake Minimization algorithm for vanilla Explainable Clustering, a kernel IMM algorithm is proposed for several kernels. __Most importantly__, a roughly $O(k^2)$ upper bound on the price of explainability holds for many kernels., when the dimension $d$ is small.

**Strengths:**

1. The paper is in general well-written, I had a good experience of reading.
2. Due to the rush I unfortunately did not read all the proof, but by a glance I think the correctness is OK. I will be happy to come back to it if any concern is raised during the discussion.
3. If this work is just combining kernelization with the decision tree framework, it is OK but not interesting enough. I like the relaxation of explainability to address this problem. More importantly, the authors give reason for doing this in Thm 1 & 2.
4. It is good to show experimental results.

**Weaknesses:**

1. The work is refrained on kernel $k$-means only, which by itself is not a limitation. But note that the line of research on explainable clustering has coupled $k$-means and median together, and the proposed algorithms, even lower bounds are similar. The $k$-center objective is indeed studied separately. It is expected that some insights should be given on how this result sheds light to $k$-median, or $k$-center even better. However I am not able to find any discussion.
2. No discussion on lower bound is involved.
3. It is based on the previous two points, I think the scope of this paper can be wider.

**Questions:**

Minor:
1. Missing related work on explainable clustering: Impossibility of Depth Reduction in Explainable Clustering
2. Related work on other notions of EXC, up to you: Optimal interpretable clustering using oblique decision trees
3. In the first work of explainable clustering, IMM also attains $O(k^2)$ upper bound but is later improved to $\tilde{O}(k)$ by balancing the mistakes with the size in the SODA paper. Have you tried this? Any thought if that would be helpful?
4. Maybe I missed it, but if not, please explain rand index.
5. Will the codes turn public?

---

> ### Author Response · Authors · 2023-11-18
> **Author response**
>
> We thank you for your review, and we are glad that you found the paper to be well-written. We appreciate that you acknowledge our strategy of relaxing the notion of explainability, which enables us to derive kernelized variants of explainable clustering. We now address the concerns raised in the weaknesses.
>
> **No relation to $k$-median and $k$-center problem are given.**
>
> The $k$-median and $k$-center problem are significantly less studied in the kernel setting. On the other hand, kernel $k$-means is closely related to spectral clustering [1] as well as clustering of (Hilbert space embeddings of) distributions [2]. The $k$-median problem requires a $\|\cdot\|_1$ norm on the RKHS, making its analysis difficult.
>
> [1] Dhillon, Inderjit S., Yuqiang Guan, and Brian Kulis. "Kernel k-means: spectral clustering and normalized cuts." Proceedings of the tenth ACM SIGKDD international conference on Knowledge discovery and data mining. 2004.
>
> [2] Vankadara, Leena C., et al. "Recovery guarantees for kernel-based clustering under non-parametric mixture models." International Conference on Artificial Intelligence and Statistics. PMLR, 2021.
>
> **No discussion on lower bound is involved.**
>
> Of course, lower bounds from the linear setting extend to the kernel setting (under a linear kernel). Nonetheless, we agree that the paper leaves some open questions (such as lower bounds for more *specific* classes of kernels such as distance-based product kernels, or other related kernel clustering problems). However, this is natural because (i) no prior works on explainable kernel clustering exist, and (ii) kernelizing existing methods is not possible directly, as Theorems 1 and 2 show, and we must hence take a broader perspective on the problem in Sections 3 and 4. This broader perspective allows us to derive rather general results and methods that are applicable to several practically relevant kernels.
>
> Thank you for pointing out additional related work, which we will add. To your question on the $O(k^2)$ bounds: We believe that improved versions of IMM (such as those obtaining $O(k \log k)$ bounds [3]) can also be applied to surrogate features.
>
> The Rand Index measures the agreement between two clustering partitions. Code will be available to the public.
>
> [3] Esfandiari, Hossein, Vahab Mirrokni, and Shyam Narayanan. "Almost tight approximation algorithms for explainable clustering." Proceedings of the 2022 Annual ACM-SIAM Symposium on Discrete Algorithms (SODA). Society for Industrial and Applied Mathematics, 2022.

---

> > ### Comment · Reviewer_8Bws · 2023-11-23
> > **Response to rebuttal**
> >
> > I thank the authors for the careful rebuttal. Please add a discussion on the lower bound and connection to other metric if possible. Also , I would encourage a better upper bound if it is a low-hanging fruit following the same trick as in [3]. Anyway, I maintain my evaluation for this work to be accepted.

---

> > > ### Author Response · Authors · 2023-11-23
> > > **Response**
> > >
> > > Due to the now ending revision period and space constraints, it is difficult to include this ad-hoc, but we will address lower bounds in the concluding discussion section. Other metrics will be in the related works section. While our methods and proofs rely on IMM, the paper will also mention possibly tighter guarantees that other algorithms could attain.

---

### Official Review · Reviewer_JcS6 · 2023-11-01

**Soundness:** 3 good
**Presentation:** 3 good
**Contribution:** 2 fair
**Rating:** 6
**Confidence:** 4

**Summary:**

The paper proposes to learn axis-aligned decision trees for kernel k-means clustering problems. Conceptually, the idea of the paper is quite similar to the work by Moshkovitz et al. (2020) but for kernel k-means clustering. First, the paper motivates for kernels that are separable over each dimension (interpretable feature maps), and theoretically shows that axis-aligned splits in the transformed feature space does not result in axis-aligned splits in the original space for the popular Gaussian kernel. By approximating kernels to achieve interpretable feature maps, the paper  then adapts existing explainable regular k-means algorithms (IMM and ExKMC) to handle the kernel k-means clustering problem. Experiments are performed on 5 datasets showing the advantage explainable kernel k-means over regular k-means.

**Strengths:**

1) As far as I know, this is the first paper addressing the problem of explainable kernel k-means clustering.
2) The paper proposes interesting approximations to the kernel function to obtain interpretable feature maps.

**Weaknesses:**

1) The paper states "interpretable variants of k-means have limited applicability in practice", but as far as I know, kernel k-means is not a widely used clustering algorithm (unlike regular k-means, spectral clustering or DBSCAN). And the need for an interpetable version of it also seems to be low.
2) The paper views the optimization problem of interpretable clustering in a two step approach: first fitting a kernel k-means and then using the greedy tree induction algorithm to fit the tree on the result of kernel k-means. But this seems to be suboptimal in that one should aim to jointly optimize the clustering objective along with the decision tree parameters.
3) The paper completely discards the approach of fitting a tree on the kernel k-means clustering labels as a supervised problem. It cites the result in Moshkovitz et al. (2020, Section 3), where an unusual Toy 2d example shown to illustrate the bad behavior which rarely occurs in practice. Fitting a classification tree (such as CART or even better algorithms such as optimal MIO-based or alternating optimization-based) on the clustering labels should produce adequate results in practice.
4) The theoretical bounds on the price of explainability seem to be quite high and not really helpful in practice. For example, does the bound of $O(dk^2)$ mean that say for MNIST the tree will do worse by (784 * 100) times than the unconstrained kernel k-means? What is the practical value of these asymptotic bounds? Experimental results seem to show that trees are doing within a constant factor the reference clustering.

**Questions:**

1) How does the results (the price of explainability) compare with tranining a CART tree on the cluster assignments?

---

> ### Author Response · Authors · 2023-11-18
> **Author response**
>
> We thank you for your review. We appreciate that you have acknowledged the novelty of the problem and our contributions to interpretable approximations of kernel functions. We hope to resolve your remaining concerns in our response.
>
> **Kernel $k$-means is not used as frequently in practice, spectral clustering or DBSCAN are preferred.**
>
> Spectral clustering is equivalent to weighted kernel $k$-means [1]. Moreover, kernel $k$-means is closely related to clustering of distributions in the RKHS [2], which connects it to DBSCAN. Understanding its explainability is therefore not only important in its own right, but also a first step to extend the existing literature on interpretable clustering to more flexible methods.
>
> [1] Dhillon, Inderjit S., Yuqiang Guan, and Brian Kulis. "Kernel k-means: spectral clustering and normalized cuts." Proceedings of the tenth ACM SIGKDD international conference on Knowledge discovery and data mining. 2004.
>
> [2] Vankadara, Leena C., et al. "Recovery guarantees for kernel-based clustering under non-parametric mixture models." International Conference on Artificial Intelligence and Statistics. PMLR, 2021.
>
> **One should jointly optimize clustering cost with the decision tree.**
>
> The recent works on explainable clustering show a clear preference towards first finding the partition, and then fitting a decision tree (for two recent analysis, see [3] and [4]). Our paper demonstrates when and how worst-case guarantees can be leveraged to the kernel setting. Joint optimization of clustering cost and decision tree is a different principle. We believe it is beyond the scope of this paper.
>
> [3] Charikar, Moses, and Lunjia Hu. "Near-optimal explainable k-means for all dimensions." Proceedings of the 2022 Annual ACM-SIAM Symposium on Discrete Algorithms (SODA). Society for Industrial and Applied Mathematics, 2022.
>
> [4] Esfandiari, Hossein, Vahab Mirrokni, and Shyam Narayanan. "Almost tight approximation algorithms for explainable clustering." Proceedings of the 2022 Annual ACM-SIAM Symposium on Discrete Algorithms (SODA). Society for Industrial and Applied Mathematics, 2022.
>
> **The paper completely discards the approach of fitting a tree on the kernel k-means clustering labels as a supervised problem.**
>
> The (Kernel) IMM algorithm is supervised, it minimizes the number of mistakes at every iteration. Similarly, the Kernel Expand algorithm can also be run on an empty tree, choosing threshold cuts that minimize the number of points assigned to a wrong cluster. For completeness, we compared CART to Kernel IMM on three clustering datasets (checking Laplace and Gaussian kernel, as well as Taylor and kernel matrix features as in the experiments). Both are similar in terms of price of explainability.
>
> | Datasets | Kernel | POE(Kernel IMM,KKM) | POE(CART, KKM) |
> | --- | --- | --- | --- |
> | Pathbased | Gaussian (kernel matrix features) | 1.06645 | 1.07004 |
> | Aggregation | Laplace (kernel matrix features) | 1.00042 | 1.00065 |
> | Flame | Gaussian (Taylor features) | 1.02256 | 1.02732 |
>
> Decision tree algorithms such as CART may perform well in practice. However, we do not know of any theoretical guarantees in the clustering regime. This is precisely what has motivated the extensive works on explainable $k$-means that this paper builds on.
>
> **The theoretical bounds on the price of explainability seem to be quite high and not really helpful in practice. For example, does the bound of  mean that say for MNIST the tree will do worse by (784 * 100) times than the unconstrained kernel k-means?**
>
> Using explainable clustering directly on images (such as MNIST) may not be meaningful because pixel values are typically not interpretable. We agree that experiments indicate that the worst-case upper bounds can be loose. It is however actually a strength of worst-case approximation guarantees that they hold for all possible data, in spite of being potentially loose for well-behaved data. Previous works on explainable $k$-means have also reported approximation ratios close to $1$ in practice (for example, see Table 1 in [5]). It is likely that algorithms tailored to specific kernels allow for tighter bounds (much in the same way as for different $\|\cdot \|_p$-norms, distinct algorithms have emerged [6]). Better guarantees could possibly be given under assumptions on the distribution, and these are interesting avenues to explore in the future. Since explainable kernel clustering has not been studied prior to our work, this paper takes a broader perspective, proposing general methods that work for a large class of kernels, regardless of the distribution.
>
> [5] Laber, Eduardo S., and Lucas Murtinho. "On the price of explainability for some clustering problems." International Conference on Machine Learning. PMLR, 2021.
>
> [6] Gamlath, Buddhima, et al. "Nearly-tight and oblivious algorithms for explainable clustering." Advances in Neural Information Processing Systems 34 (2021): 28929-28939.

---

> > ### Comment · Reviewer_JcS6 · 2023-11-19
> >
> > I thank the authors for responding to my concerns. I raise my score. I recommend the authors to include the CART results in the main paper to show how a simple approach performs well in practice despite not having theoretical guarantees.

---

### Author Response · Authors · 2023-11-22
**Revision**

We thank the reviewers, and have updated the paper to incorporate minor changes suggested in the reviews. These include additional related works, fixing of typos, a comparison with CART (noted in the main paper, included in the appendix), and a slight reordering of the sections in the appendix.

---

### Meta-Review · Area_Chair_jG1n · 2023-12-07

**Metareview:**

All reviewers were positive about the paper and I concur with them. It extends previous results on interpretable clustering to the kernel clustering case.

There were several important issues noted by the reviewers, which the authors should address/discuss in the final version. In particular, one important shortcoming of interpretable clustering using axis-aligned trees is that they are usually a poor model of the clustering regions. The paper avoids this by using axis-aligned trees in the transformed space. A more direct way is via sparse oblique trees, as proposed in this reference mentioned in the reviewer discussion: "Optimal interpretable clustering using oblique decision trees", KDD 2022. This shows that k-means defines a partition that can be seen exactly as an oblique tree; and how this tree can be further optimized for interpretability, by reducing the number of nodes or making the hyperplanes sparse.

**Justification For Why Not Higher Score:**

Results of somewhat narrow interest.

**Justification For Why Not Lower Score:**

Relevant contribution to the body of work on interpretable clustering.

---

### Decision · Program_Chairs · 2024-01-16

Accept (poster)